# Robust Subspace Approximation in a Stream

**Roie Levin**[1]
roiel@cs.cmu.edu

**Anish Sevekari**[2]
asevekar@andrew.cmu.edu

**David P. Woodruff**[1]
dwoodruf@cs.cmu.edu

[1] Computer Science Department, Carnegie Mellon University, Pittsburgh, PA 15213
[2] Department of Mathematical Sciences, Carnegie Mellon University, Pittsburgh, PA 15213

## Abstract

We study robust subspace estimation in the streaming and distributed settings. Given a set of $n$ data points $\{a_i\}_{i=1}^n$ in $\mathbb{R}^d$ and an integer $k$, we wish to find a linear subspace $S$ of dimension $k$ for which $\sum_i M(\text{dist}(S, a_i))$ is minimized, where $\text{dist}(S, x) := \min_{y \in S} \|x - y\|_2$, and $M(\cdot)$ is some loss function. When $M$ is the identity function, $S$ gives a subspace that is more robust to outliers than that provided by the truncated SVD. Though the problem is NP-hard, it is approximable within a $(1 + \epsilon)$ factor in polynomial time when $k$ and $\epsilon$ are constant. We give the first sublinear approximation algorithm for this problem in the turnstile streaming and arbitrary partition distributed models, achieving the same time guarantees as in the offline case. Our algorithm is the first based entirely on oblivious dimensionality reduction, and significantly simplifies prior methods for this problem, which held in neither the streaming nor distributed models.

## 1 Introduction

A fundamental problem in large-scale machine learning is that of subspace approximation. Given a set of $n$ data points $\{a_i\}_{i=1}^n$ in $\mathbb{R}^d$ and an integer $k$, we wish to find a linear subspace $S$ of dimension $k$ for which $\sum_i M(\text{dist}(S, a_i))$ is minimized, where $\text{dist}(S, x) := \min_{y \in S} \|x - y\|_2$, and $M(\cdot)$ is some loss function. When $M(\cdot) = (\cdot)^2$, this is the well-studied least squares subspace approximation problem. The minimizer in this case can be computed exactly by computing the truncated SVD of the data matrix.

Otherwise $M$ is often chosen from $(\cdot)^p$ for some $p \geq 0$, or from a class of functions called $M$-estimators, with the goal of providing a more robust estimate than least squares in the face of outliers. Indeed, for $p < 2$, since one is not squaring the distances to the subspace, one is placing less emphasis on outliers and therefore capturing more of the remaining data points. For example, when $M$ is the identity function, we are finding a subspace so as to minimize the sum of distances to it, which could arguably be more natural than finding a subspace so as to minimize the sum of squared distances. We can write this problem in the following form:

$$\min_{S \dim k} \sum_i \text{dist}(S, a_i) = \min_{X \text{ rank } k} \sum_i \|(A - AX)_{i*}\|_2$$

where $A$ is the matrix in which the $i$-th row is the vector $a_i$. This is the form of robust subspace approximation that we study in this work. We will be interested in the approximate version of the problem for which the goal is to output a $k$-dimensional subspace $S'$ for which with high probability,

$$\sum_i \text{dist}(S', a_i) \leq (1 + \epsilon) \sum_i \text{dist}(S, a_i) \tag{1}$$

The particular form with $M$ equal to the identity was introduced to the machine learning community by Ding et al. [10], though these authors employed heuristic solutions. The series of work in

[7],[15] and [8, 12, 20, 5] shows that if $M(\cdot) = |\cdot|^p$ for $p \neq 2$, there is no algorithm that outputs a $(1 + 1/\operatorname{poly}(d))$ approximation to this problem unless $P = NP$. However, [5] also show that for any $p$ there is an algorithm that runs in $O(\operatorname{nnz}(A) + (n + d)\operatorname{poly}(k/\epsilon) + \exp(\operatorname{poly}(k/\epsilon)))$ time and outputs a $k$-dimensional subspace whose cost is within a $(1 + \epsilon)$ factor of the optimal solution cost. This provides a considerable computational savings since in most applications $k \ll d \ll n$. Their work builds upon techniques developed in [13] and [11] which give $O\left(nd \cdot \operatorname{poly}(k/\epsilon) + \exp\left((k/\epsilon)^{O(p)}\right)\right)$ time algorithms for the $p \geq 1$ case. These in turn build on the weak coreset construction of [9]. In other related work [6] give algorithms for performing regression with a variety of $M$-estimator loss functions.

**Our Contributions.**    We give the first sketching-based solution to this problem. Namely, we show it suffices to compute $Z \cdot A$, where $Z$ is a $\operatorname{poly}(\log(nd)k\epsilon^{-1}) \times n$ random matrix with entries chosen obliviously to the entries of $A$. The matrix $Z$ is a block matrix with blocks consisting of independent Gaussian entries, while other blocks consist of independent Cauchy random variables, and yet other blocks are sparse matrices with non-zero entries in $\{-1, 1\}$. Previously such sketching-based solutions were known only for $M(\cdot) = (\cdot)^2$. Prior algorithms [8, 12, 20, 5] also could not be implemented as single-shot sketching algorithms since they require first making a pass over the data to obtain a crude approximation, and then using (often adaptive) sampling methods in future passes to refine to a $(1 + \epsilon)$-approximation. Our sketching-based algorithm, achieving $O(\operatorname{nnz}(A) + (n + d)\operatorname{poly}(\log(nd)k/\epsilon) + \exp(\operatorname{poly}(k\epsilon^{-1})))$ time, matches the running time of previous algorithms and has considerable benefits as described below.

*Streaming Model.* Since $Z$ is linear and oblivious, one can maintain $Z \cdot A$ in the presence of insertions and deletions to the entries of $A$. Indeed, given the update $A_{i,j} \leftarrow A_{i,j} + \Delta$ for some $\Delta \in \mathbb{R}$, we simply update the $j$-th column $ZA_j$ in our sketch to $ZA_j + \Delta \cdot Z \cdot e_i$, where $e_i$ is the $i$-th standard unit vector. Also, the entries of $Z$ can be represented with limited independence, and so $Z$ can be stored with a short random seed. Consequently, we obtain the first algorithm with $d \operatorname{poly}(\log(nd)k\epsilon^{-1})$ memory for this problem in the standard turnstile data stream model [19]. In this model, $A \in \mathbb{R}^{n \times d}$ is initially the zero matrix, and we receive a stream of updates to $A$ where the $i$-th update is of the form $(x_i, y_i, c_i)$, which means that $A_{x_i, y_i}$ should be incremented by $c_i$. We are allowed one pass over the stream, and should output a rank-$k$ matrix $X'$ which is a $(1 + \epsilon)$ approximation to the robust subspace estimation problem, namely $\sum_i \|(A - AX')_{i*}\|_2 \leq (1 + \epsilon)\min_{X \text{ rank } k} \sum_i \|(A - AX)_{i*}\|_2$. The space complexity of the algorithm is the total number of words required to store this information during the stream. Here, each word is $O(\log(nd))$ bits. Our algorithm achieves $d \operatorname{poly}(\log(nd)k\epsilon^{-1})$ memory, and so only logarithmically depends on $n$. This is comparable to the memory of streaming algorithms when $M(\cdot) = (\cdot)^2$ [3, 14], which is the only prior case for which streaming algorithms were known.

*Distributed Model.*   Since our algorithm maintains $Z \cdot A$ for an oblivious linear sketch $Z$, it is parallelizable, and can be used to solve the problem in the distributed setting in which there are $s$ machines holding $A^1, A^2, \ldots, A^s$, respectively, and $A = \sum_{i=1}^{s} A^i$. This is called the *arbitrary partition model* [17]. In this model, we can solve the problem in one round with $s \cdot d \operatorname{poly}(\log(nd)k\epsilon^{-1})$ communication by having each machine agree upon (a short seed describing) $Z$, and sending $ZA^i$ to a central coordinator who computes and runs our algorithm on $Z \cdot A = \sum_i ZA^i$. The arbitrary partition model is stronger than the so-called row partition model, in which the points (rows of $A$) are partitioned across machines. For example, if each machine corresponds to a shop, the rows of $A$ correspond to customers, the columns of $A$ correspond to items, and $A^i_{c,d}$ indicates how many times customer $c$ purchased item $d$ at shop $i$, then the row partition model requires customers to make purchases at a single shop. In contrast, in the arbitrary partition model, customers can purchase items at multiple shops.

## 2   Notation and Terminology

For a matrix $A$, let $A_{i*}$ denote the $i$-th row of $A$, and $A_{*j}$ denote the $j$-th column of $A$.

**Definition 2.1** ($\|\cdot\|_{2,1}, \|\cdot\|_{1,2}, \|\cdot\|_{1,1}, \|\cdot\|_{\mathrm{med},1}, \|\cdot\|_F$)**.** *For a matrix $A \in \mathbb{R}^{n \times m}$, let:*

$$\|A\|_{2,1} \equiv \sum_i \|A_{i*}\|_2 \qquad \|A\|_{1,2} \equiv \|A^\intercal\|_{2,1} = \sum_j \|A_{*j}\|_2$$

$$\|A\|_F \equiv \sqrt{\sum_i \|A_{i*}\|_2^2} \qquad \|A\|_{1,1} \equiv \sum_i \|A_{i*}\|_1 \qquad\qquad \|A\|_{\mathrm{med},1} \equiv \sum_j \|A_{*j}\|_{\mathrm{med}}$$

*where $\|\cdot\|_{\mathrm{med}}$ denotes the function that takes the median of absolute values.*

**Definition 2.2** ($X^*, \Delta^*$)**.** *Let:*

$$\Delta^* \equiv \min_{X \text{ rank } k} \|A - AX\|_{2,1} \qquad\qquad X^* \equiv \operatorname*{argmin}_{X \text{ rank } k} \|A - AX\|_{2,1}$$

**Definition 2.3** (($\alpha, \beta$)-coreset)**.** *For a matrix $A \in \mathbb{R}^{n \times d}$ and a target rank $k$, $W$ is an $(\alpha, \beta)$-coreset if its row space is an $\alpha$-dimensional subspace of $\mathbb{R}^d$ that contains a $\beta$-approximation to $X^*$. Formally:*

$$\operatorname*{argmin}_{X \text{ rank } k} \|A - AXW\|_{2,1} \leq \beta \Delta^*$$

**Definition 2.4** (Count-Sketch Matrix)**.** *A random matrix $S \in \mathbb{R}^{r \times t}$ is a Count-Sketch matrix if it is constructed via the following procedure. For each of the $t$ columns $S_{*i}$, we first independently choose a uniformly random row $h(i) \in \{1, 2, ..., r\}$. Then, we choose a uniformly random element of $\{-1, 1\}$ denoted $\sigma(i)$. We set $S_{h(i),i} = \sigma(i)$ and set $S_{j,i} = 0$ for all $j \neq i$.*

For the applications of Count-Sketch matrices in this paper, it suffices to use $O(1)$-wise instead of full independence for the hash and sign functions. Thus these can be stored in $O(1)$ space, and multiplication $SA$ can be computed in $\mathrm{nnz}(A)$ time. For more background on such sketching matrices, we refer the reader to the monograph [22].

We also use the following notation: $[n]$ denotes the set $\{1, 2, 3, \cdots n\}$. $[\![E]\!]$ denotes the indicator function for event $E$. $\mathrm{nnz}(A)$ denotes the number of non-zero entries of $A$. $A^-$ denotes the pseudoinverse of $A$. $\mathcal{I}$ denotes the identity matrix.

## 3   Algorithm Overview

At a high level we follow the framework put forth in [5] which gives the first input sparsity time algorithm for the robust subspace approximation problem. In their work Clarkson and Woodruff first find a crude $(\mathrm{poly}(k), K)$-coreset for the problem. They then use a non-adaptive implementation of a residual sampling technique from [9] to improve the approximation quality but increase the dimension, yielding a $(K \mathrm{poly}(k), 1 + \epsilon)$-coreset. From here they further use dimension reducing sketches to reduce to an instance with parameters that depend only polynomially on $k/\epsilon$. Finally they pay a cost exponential only in $\mathrm{poly}(k/\epsilon)$ to solve the small problem via a black box algorithm of [2].

There are several major obstacles to directly porting this technique to the streaming setting. For one, the construction of the crude approximation subspace uses leverage score sampling matrices which are non-oblivious and thus not usable in 1-pass turnstile model algorithms. We circumvent this difficulty in Section 4.1 by showing that if $T$ is a sparse $\mathrm{poly}(k) \times n$ matrix of Cauchy random variables, the row span of $TA$ contains a rank-$k$ matrix which is a $\log(d) \mathrm{poly}(k)$ approximation to the best rank-$k$ matrix under the $\|\cdot\|_{2,1}$ norm.

Second, the residual sampling step requires sampling rows of $A$ with respect to probabilities proportional to their distance to the crude approximation (in our case $TA$). This is challenging because one does not know $TA$ until the end of the stream, much less the distances of rows of $A$ to $TA$. We handle this in Section 4.2 using a row-sampling data structure of [18] developed for regression, which for a matrix $B$ maintains a sketch $HB$ in a stream from which one can extract samples of rows of $B$ according to probabilities given by their norms. By linearity, it suffices to maintain $HA$ and $TA$ in parallel in the stream, and apply the sample extraction procedure to $HA \cdot (\mathcal{I} - P_{TA})$, where $P_{TA} = (TA)^\intercal (TA(TA)^\intercal)^{-1} TA$ is the projection onto the rowspace of $TA$. Unfortunately, the extraction procedure only returns noisy perturbations of the original rows which majorly invalidates the analysis in [5] of the residual sampling. In Section 4.2 we give an analysis of non-adaptive noisy

residual sampling which we name BOOTSTRAPCORESET. This gives a procedure for transforming our $\text{poly}(k)$-dimensional space containing a $\text{poly}(k)\log(d)$ approximation into a $\text{poly}(k)\log(d)$-dimensional space containing a $3/2$ factor approximation.

Third, requiring the initial crude approximation to be oblivious yields a coarser $\log(d)\,\text{poly}(k)$ initial approximation than the constant factor approximation of [5]. Thus the dimension of the subspace after residual sampling is $\text{poly}(k)\log(d)$. Applying dimension reduction techniques reduces the problem to an instance with $\text{poly}(k)$ rows and $\log(d)\,\text{poly}(k)$ columns. Here the black box algorithm of [2] would take time $d^{\text{poly}(k)}$ which is no longer fixed parameter tractable as desired. Our key insight is that finding the best rank-$k$ matrix under the Frobenius norm, which can be done efficiently, is a $\sqrt{\log d}(\log\log d)\,\text{poly}(k)$ approximation to the $\|\cdot\|_{2,1}$ norm minimizer. From here we can repeat the residual sampling argument which this time yields a small instance with $\text{poly}(k)$ rows by $\sqrt{\log d}(\log\log d)\,\text{poly}(k/\epsilon)$ columns. Sublogarithmic in $d$ makes all the difference and now enumerating can be done in time $(n+d)\,\text{poly}(k/\epsilon) + \exp(\text{poly}(k/\epsilon))$. All this is done in parallel in a single pass of the stream.

Lastly, the sketching techniques applied after the residual sampling are not oblivious in [5]. We instead use an obvlious median based embedding in Section 5.1, and show that we can still use the black box algorithm of [2] to find the minimizer under the $\|\cdot\|_{\text{med},1}$ norm in Section 5.2.

We present our results as two algorithms for the robust subspace approximation problem. The first runs in fully polynomial time but gives a coarse approximation guarantee, which corresponds to stopping before repeating the residual sampling a second time. The second algorithm captures the entire procedure, and uses the first as a subroutine.

---

**Algorithm 1** COARSEAPPROX

---

**Input:** $A \in \mathbb{R}^{n \times d}$ as a stream
**Output:** $X \in \mathbb{R}^{d \times d}$ such that $\|A - AX\|_{2,1} \le \sqrt{\log d}(\log\log d)\,\text{poly}(k)\Delta^*$

1: $T \in \mathbb{R}^{\text{poly}(k) \times n} \leftarrow$ Sparse Cauchy matrix // as in Thm. 4.1
2: $C_1 \in \mathbb{R}^{\text{poly}(k) \times n} \leftarrow$ Sparse Cauchy matrix // as in Thm. 4.4
3: $S_1 \in \mathbb{R}^{\log d \cdot \text{poly}(k) \times d} \leftarrow$ Count Sketch composed with Gaussian // as in Thm. 4.3
4: $R_1 \in \mathbb{R}^{\text{poly}(k) \times d} \leftarrow$ Count Sketch matrix // as in Thm. 4.3
5: $G_1 \in \mathbb{R}^{\log d \cdot \text{poly}(k) \times \log d \cdot \text{poly}(k)} \leftarrow$ Gaussian matrix // as in Thm. 4.4
6: Compute $TA$ online
7: Compute $C_1A$ online
8: $U_1^\intercal \in \mathbb{R}^{\log d\,\text{poly}(k) \times d} \leftarrow$ BOOTSTRAPCORESET$(A, TA, 1/2)$ // as in Alg. 3
9: $\hat{X} \in \mathbb{R}^{\text{poly}(k) \times \log d\,\text{poly}(k)} \leftarrow \text{argmin}_{X \text{ rank } k} \|C_1(A - AR_1^\intercal X U_1^\intercal)S_1^\intercal G_1\|_F$ // as in Fact 4.2
10: **return** $R_1^\intercal \hat{X} U^\intercal$

---

**Theorem 3.1** (Coarse Approximation in Polynomial Time). *Given a matrix $A \in R^{n \times d}$, Algorithm 1 with constant probability computes a rank $k$ matrix $X \in \mathbb{R}^{d \times d}$ such that:*

$$\|A - AX\|_{2,1} \le \sqrt{\log d}(\log\log d) \cdot \text{poly}(k) \cdot \|A - AX^*\|_{2,1}$$

*that runs in time $O(nnz(A)) + d\,\text{poly}(k\log(nd))$. Furthermore, it can be implemented as a one-pass streaming algorithm with space $O\left(d\,\text{poly}(k\log(nd))\right)$ and time per update $O(\text{poly}(\log(nd)k))$.*

**Proof Sketch** We show the following are true in subsequent sections:

1. The row span of $TA$ is a $(\text{poly}(k), \log d \cdot \text{poly}(k))$-coreset for $A$ (Section 4.1) with probability $24/25$.

2. BOOTSTRAPCORESET$(A, TA, 1/2)$ is a $(\log d \cdot \text{poly}(k), 3/2)$-coreset with probability $49/50$ (Section 4.2).

3. If:
$$\hat{X} = \underset{X \text{ rank } k}{\text{argmin}} \|C_1 A S_1^\intercal G_1 - C_1 A R_1^\intercal X U_1^\intercal S_1^\intercal G_1\|_F$$
then with probability $47/50$:
$$\left\|A - AR_1^\intercal \hat{X} U_1^\intercal\right\|_{2,1} \le \text{poly}(k)\sqrt{\log d}\log\log d \cdot \Delta^*$$

(Sections 4.3 and 4.4, with $\epsilon = 1/2$).

By a union bound, with probability 88/100 all the statements above hold, and the theorem is proved. BOOTSTRAPCORESET requires $d \operatorname{poly}(k \log(nd))$ space and time. Left matrix multiplications by Sparse Cauchy matrices $TA$ and $C_1 A$ can be done in $O(\operatorname{nnz}(A))$ time (see Section J of [21] for a full description of Sparse Cauchy matrices). Computing remaining matrix products and $\hat{X}$ requires time $d \operatorname{poly}(k \log d)$. $\qquad\square$

---

**Algorithm 2** $(1 + \epsilon)$-APPROX

---

**Input:** $A \in \mathbb{R}^{n \times d}$ as a stream
**Output:** $X \in \mathbb{R}^{d \times d}$ such that $\|A - AX\|_{2,1} \le (1 + \epsilon)\Delta^*$

1: $\hat{X} \in \mathbb{R}^{\operatorname{poly}(k) \times \log d \operatorname{poly}(k)} \leftarrow$ COARSEAPPROX$(A)$ // as in Thm. 3.1
2: $C_2 \in \mathbb{R}^{\sqrt{\log d}(\log \log d) \operatorname{poly}(k/\epsilon) \times n} \leftarrow$ Cauchy matrix // as in Thm. 5.1
3: $S_2 \in \mathbb{R}^{\sqrt{\log d}(\log \log d) \cdot \operatorname{poly}(k/\epsilon) \times d} \leftarrow$ Count Sketch composed with Gaussian // as in Thm. 4.3
4: $R_2 \in \mathbb{R}^{\operatorname{poly}(k/\epsilon) \times d} \leftarrow$ Count Sketch matrix // as in Thm. 4.3
5: $G_2 \in \mathbb{R}^{\sqrt{\log d}(\log \log d) \cdot \operatorname{poly}(k/\epsilon) \times \sqrt{\log d}(\log \log d) \cdot \operatorname{poly}(k/\epsilon)} \leftarrow$ Gaussian matrix // as in Thm. 5.1
6: Compute $AR_2^{\mathsf{T}}$ online
7: Compute $AS_2^{\mathsf{T}}$ online
8: Let $V \in \mathbb{R}^{\log d \operatorname{poly}(k) \times k}$ be such that $\hat{X} = WV^{\mathsf{T}}$ is the rank-$k$ decomposition of $\hat{X}$
9: $U_2^{\mathsf{T}} \in \mathbb{R}^{\operatorname{poly}(k/\epsilon)\sqrt{\log d} \log \log d \times d} \leftarrow$ BOOTSTRAPCORESET$(A, V^{\mathsf{T}}U_1^{\mathsf{T}}, \epsilon')$ // as in Alg. 3, $U_1$ as computed during COARSEAPPROX in line 1.
10: $\hat{X}' \in \mathbb{R}^{\operatorname{poly}(k/\epsilon) \times \operatorname{poly}(k/\epsilon)\sqrt{\log d} \log \log d} \leftarrow \operatorname{argmin}_{X \text{ rank } k} \|C_2(A - AR_2^{\mathsf{T}}XU_2^{\mathsf{T}})S_2^{\mathsf{T}}G_2\|_{\text{med},1}$ // as in Thm. 5.2
11: **return** $R_2^{\mathsf{T}}\hat{X}'U'^{\mathsf{T}}$

---

**Theorem 3.2** ($(1 + \epsilon)$-Approximation). *Given a matrix $A \in R^{n \times d}$, Algorithm 2 with constant probability computes a rank $k$ matrix $X \in \mathbb{R}^{d \times d}$ such that:*

$$\|A - AX\|_{2,1} \le (1 + \epsilon)\|A - AX^*\|_{2,1}$$

*that runs in time*

$$O(nnz(A)) + (n + d)\operatorname{poly}\left(\frac{k \log(nd)}{\epsilon}\right) + \exp\left(\operatorname{poly}\left(\frac{k}{\epsilon}\right)\right)$$

*Furthermore, it can be implemented as a one-pass streaming algorithm with space $O\left(d \operatorname{poly}\left(\frac{k \log(nd)}{\epsilon}\right)\right)$ and time per update $O(\operatorname{poly}(\log(nd)k/\epsilon))$.*

**Proof Sketch** We show the following are true in subsequent sections:

1. If $V$ is such that $\hat{X} = WV^{\mathsf{T}}$, then $V^{\mathsf{T}}$ is a $(\operatorname{poly}(k), \operatorname{poly}(k)\sqrt{\log d} \log \log d)$-coreset with probability 88/100 (Theorem 3.1).

2. BOOTSTRAPCORESET$(A, V^{\mathsf{T}}U_1^{\mathsf{T}}, \epsilon')$ is a $(\operatorname{poly}(k/\epsilon')\sqrt{\log d} \log \log d, (1 + \epsilon'))$-coreset with probability 49/50 (Reusing Section 4.2).

3. If:
$$\hat{X}' \leftarrow \operatorname*{argmin}_{X} \|C_2(A - AR_2^{\mathsf{T}}XU_2^{\mathsf{T}})S_2^{\mathsf{T}}G_2\|_{\text{med},1}$$

   then with probability 19/20:

$$\left\|A - AR_2^{\mathsf{T}}\hat{X}'U_2^{\mathsf{T}}\right\|_{2,1} \le (1 + O(\epsilon'))\Delta^*$$

   (Reusing Section 4.3 and Section 5.1).

4. A black box algorithm of [2] computes $\hat{X}'$ to within $(1 + O(\epsilon'))$ (Section 5.2).

By a union bound, with probability 81/100 all the statements above hold. Setting $\epsilon'$ appropriately small as a function of $\epsilon$, the theorem is proved.

COARSEAPPROX and BOOTSTRAPCORESET together require $d \operatorname{poly}(k \log(nd)/\epsilon)$ space and $O(\operatorname{nnz}(A)) + d \operatorname{poly}(k \log(nd)/\epsilon)$ time. Right multiplication by the sketching matrices $AS_2^{\mathsf{T}}$ and $AR_2^{\mathsf{T}}$ can be done in time $\operatorname{nnz}(A)$. Computing remaining matrix products and $\hat{X}'$ requires time $(n+d) \operatorname{poly}(\log(d)k/\epsilon) + \exp(\operatorname{poly}(k/\epsilon))$ (See end of Section 5.2 for details on this last bound). □

We give further proofs and details of these theorems in subsequent sections. Refer to the full version of the paper for complete proofs.

# 4 Coarse Approximation

## 4.1 Initial Coreset Construction

We construct a $(\operatorname{poly}(k), \log d \cdot \operatorname{poly}(k))$-coreset which will serve as our starting point.

**Theorem 4.1.** *If $T \in \mathbb{R}^{\operatorname{poly}(k) \times n}$ is a Sparse Cauchy matrix, then the row space of $TA$ contains a $k$ dimensional subspace with corresponding projection matrix $X'$ such that with probability $24/25$:*

$$\|A - AX'\|_{2,1} \leq \log d \cdot \operatorname{poly}(k) \min_{X \ rank \ k} \|A - AX\|_{2,1} = \log d \cdot \operatorname{poly}(k) \cdot \Delta^*$$

In order to deal with the awkward $\|\cdot\|_{2,1}$ norm, here and several times elsewhere we make use of a well known theorem due to Dvoretzky to convert it into an entrywise 1-norm.

**Fact 4.1** (Dvoretzky's Theorem (Special Case), Section 3.3 of [16])**.** *There exists an appropriately scaled Gaussian Matrix $G \in \mathbb{R}^{d \times \frac{d \log(1/\epsilon)}{\epsilon^2}}$ such that w.h.p. the following holds for all $y \in \mathbb{R}^d$ simultaneously*

$$\|y^{\mathsf{T}}G\|_1 \in (1 \pm \epsilon) \|y^{\mathsf{T}}\|_2$$

Thus the rowspace of $TA$ with $T$ as in Theorem 4.1 above is a $(\operatorname{poly}(k), \log d \cdot \operatorname{poly}(k))$-coreset for $A$.

## 4.2 Bootstrapping a Coreset

Given a poor coreset $Q$ for $A$, we now show how to leverage known results about residual sampling from [9] and [5] to obtain a better coreset of slightly larger dimension.

---

**Algorithm 3** BOOTSTRAPCORESET

**Input:** $A \in \mathbb{R}^{n \times d}$, $Q \in \mathbb{R}^{\alpha \times d}$ $(\alpha, \beta)$-coreset, $\epsilon \in (0, 1)$
**Output:** $U \in \mathbb{R}^{(\alpha + \beta \operatorname{poly}(k/\epsilon)) \times d}$ $(\alpha + \beta \operatorname{poly}(k/\epsilon), (1+\epsilon))$-coresets
1: Compute $HA$ online // as in Lem. 4.2.2
2: $P \leftarrow \beta \operatorname{poly}(k/\epsilon)$ samples of rows of $A(\mathcal{I} - Q)$ according to $\mathcal{P}(HA(\mathcal{I} - Q))$ // as in Lem. 4.2.2
3: $U^{\mathsf{T}} \leftarrow$ Orthonormal basis for $\operatorname{RowSpan}\left( \begin{bmatrix} Q \\ P \end{bmatrix} \right)$
4: **return** $U^{\mathsf{T}}$

---

**Theorem 4.2.** *Given $Q$, an $(\alpha, \beta)$-coreset for $A$, with probability $49/50$ BOOTSTRAPCORESET returns an $(\alpha + \beta \operatorname{poly}(k/\epsilon), (1+\epsilon))$-coreset for $A$. Furthermore BOOTSTRAPCORESET runs in space and time $O(d \operatorname{poly}(\beta \log(nd)k/\epsilon))$, with $\operatorname{poly}(\beta \log(nd)k/\epsilon)$ time per update in the streaming setting.*

*Proof.* Consider the following idealized noisy sampling process that samples rows of a matrix $B$. Sample a row $B_i$ of $B$ with probability $\frac{\|B_i\|_2}{\|B\|_{2,1}}$ and add an arbitrary noise vector $E_i$ such that $\|E_i\|_2 \leq \nu \|B_i\|_{2,1}$, where we fix the parameter $\nu = \frac{\epsilon}{100k\beta}$. Supposing we had such a process $\mathcal{P}^*(B)$, we can prove the following lemma.

**Lemma 4.2.1.** *Suppose $Q$ is an $(\alpha, \beta)$-coreset for $A$, and $P$ is a noisy subset of rows of the residual $A(\mathcal{I} - Q)$ of size $\beta(\operatorname{poly} k/\epsilon)$ each sampled according to $\mathcal{P}^*(A(\mathcal{I} - Q))$. Then with probability*

99/100, $\mathrm{RowSpan}(Q) \cup \mathrm{RowSpan}(P)$ *is an* $(\alpha + \beta \operatorname{poly}(k/\epsilon))$ *dimensional subspace containing a $k$-dimensional subspace with corresponding projection matrix $X'$ such that:*

$$\|A - AX'\|_{2,1} \leq (1 + \epsilon)\Delta^*$$

It remains to show that we can sample from $\mathcal{P}^*$ in a stream.

**Lemma 4.2.2.** *Let $B \in \mathbb{R}^{n \times d}$ be a matrix, and let $\delta, \nu \in (0, 1)$ be given. Also let $s$ be a given integer. Then there is an oblivious sketching matrix $H \in \mathbb{R}^{\operatorname{poly}(s/(\delta\nu)) \times n}$ and a sampling process $\mathcal{P}$, such that $\mathcal{P}(HB)$ returns a collection of $s' = O(s)$ distinct row indices $i_1, \ldots, i_{s'} \in [n]$ and approximations $\tilde{B}_{i_j} = B_{i_j} + E_{i_j}$ with $\|E_{i_j}\|_2 \leq \nu \cdot \|B_{i_j}\|_2$ for $j = 1, \ldots, s$. With probability $1 - \delta$ over the choice of $H$, the probability an index $i$ appears in the sampled set $\{i_1, \ldots, i_{s'}\}$ is at least the probability that $i$ appears in a set of $s$ samples without replacement from the distribution $\left(\frac{\|B_{1,*}\|_2}{\|B\|_{2,1}}, \ldots \frac{\|B_{n,*}\|_2}{\|B\|_{2,1}}\right)$. Furthermore the multiplication $HB$ and sampling process $\mathcal{P}$ can be done in $\operatorname{nnz}(B) + d \cdot \operatorname{poly}(s/(\delta\nu))$ time, and can be implemented in the streaming model with $d \cdot \operatorname{poly}(s/(\delta\nu))$ bits of space.*

Setting $b = \log(nd)$, $\delta = 1/100$, $\gamma = \nu = \frac{\epsilon}{100k\beta}$ and $s = \beta \operatorname{poly}(k/\epsilon)$, it follows that $P$ contains $\beta \operatorname{poly}(k/\epsilon)$ samples from $\mathcal{P}^*(A(\mathcal{I} - Q))$ with probability 99/100. By Lemma 4.2.1 and a union bound, the projection matrix of $\mathrm{RowSpan}(Q) \cup \mathrm{RowSpan}(P)$ is an $(\alpha + \beta \operatorname{poly}(k/\epsilon), (1 + \epsilon))$-coreset for $A$ with probability 49/50. BOOTSTRAPCORESET takes total time $O(\operatorname{nnz}(A)) + O(d \operatorname{poly}(\beta \log(nd)k/\epsilon))$ and space $O(d \operatorname{poly}(\beta \log(nd)k/\epsilon))$. $\qquad\square$

Note that in our main algorithm we cannot compute the projection $A(\mathcal{I} - Q)$ until the after the stream is finished. Fortunately, since $H$ is oblivious, we can right multiply $HA$ by $(\mathcal{I} - Q)$ once $Q$ is available, and only then perform the sampling procedure $\mathcal{P}$.

## 4.3 Right Dimension Reduction

We show how to reduce the right dimension of our problem. This result is used in both Algorithm 1 and Algorithm 2.

**Theorem 4.3.** *If $U^\mathsf{T}$ is an $(\alpha, \beta)$-coreset, $S \in \mathbb{R}^{\alpha \cdot \operatorname{poly}(k/\epsilon) \times d}$ is a CountSketch matrix composed with a matrix of i.i.d. Gaussians, and $R \in \mathbb{R}^{d \times \operatorname{poly}(k/\epsilon)}$ is a CountSketch matrix, then with probability 49/50, if $X' = \operatorname{argmin}_X \|AS^\mathsf{T} - AR^\mathsf{T}XU^\mathsf{T}S^\mathsf{T}\|_{2,1}$ then:*

$$\|A - AR^\mathsf{T}X'U^\mathsf{T}\|_{2,1} \leq (1 + O(\epsilon)) \min_{X \text{ rank } k} \|A - AXU^\mathsf{T}\|_{2,1}$$

## 4.4 Left Dimension Reduction

We show how to reduce the left dimension of our problem. Together with results from Section 4.3, this preserves the solution to $X^*$ to within a coarse $\sqrt{\log d} \log\log d \cdot \operatorname{poly}(k/\epsilon)$ factor.

**Theorem 4.4.** *Suppose the matrices $S_1$, $R_1$ and $U_1$ are as in Algorithm 1. If $C_1 \in \mathbb{R}^{\operatorname{poly}(k/\epsilon) \times n}$ is a Sparse Cauchy matrix, and $G_1 \in \mathbb{R}^{\log d \operatorname{poly}(k/\epsilon) \times \log d \operatorname{poly}(k/\epsilon)}$ is a matrix of appropriately scaled i.i.d. Gaussians (as in Fact 4.1), and*

$$\hat{X} = \operatorname*{argmin}_{X \text{ rank } k} \|C_1 AS_1^\mathsf{T}G_1 - C_1 AR_1^\mathsf{T}XU_1^\mathsf{T}S_1^\mathsf{T}G_1\|_F$$

*then with probability 24/25:* $\left\|AS_1^\mathsf{T} - AR_1^\mathsf{T}\hat{X}U_1^\mathsf{T}S_1^\mathsf{T}\right\|_{2,1} \leq \sqrt{\log d} \log\log d \cdot \operatorname{poly}(k/\epsilon) \cdot \Delta^*$

The rank constrained Frobenius norm minimization problem above has a closed form solution.

**Fact 4.2.** *For a matrix $M$, let $U_M \Sigma_M V_M^\mathsf{T}$ be the SVD of $M$. Then:*

$$\operatorname*{argmin}_{X \text{ rank } k} \|Y - ZXW\|_F = Z^-[U_Z U_Z^\mathsf{T} Y V_W V_W^\mathsf{T}]_k W^-$$

# 5 $(1 + \epsilon)$-Approximation

## 5.1 Left Dimension Reduction

The following median based embedding allows us to reduce the left dimension of our problem. Together with results from Section 4.3, this preserves the solution to $X^*$ to within a $(1 + O(\epsilon))$ factor.

**Theorem 5.1.** *Suppose $S_2$, $R_2$ and $U_2$ are as in Algorithm 2. If $C_2 \in \mathbb{R}^{\sqrt{\log d} \log \log d \operatorname{poly}(k/\epsilon) \times n}$ is a Cauchy matrix, and $G_2 \in \mathbb{R}^{\sqrt{\log d} \log \log d \operatorname{poly}(k/\epsilon) \times \sqrt{\log d} \log \log d \operatorname{poly}(k/\epsilon)}$ is a matrix of appropriately scaled i.i.d. Gaussians (as in Fact 4.1), and:*

$$\hat{X}' = \operatorname*{argmin}_{X \text{ rank } k} \|C_2 A S_2^\mathsf{T} G_2 - C_2 A R_2^\mathsf{T} X U_2^\mathsf{T} S_2^\mathsf{T} G_2\|_{\mathrm{med},1}$$

*then with probability $99/100$:*

$$\left\| A S_2^\mathsf{T} G_2 - A R_2^\mathsf{T} \hat{X}' U_2^\mathsf{T} S_2^\mathsf{T} G_2 \right\|_{1,1} \le (1 + \epsilon) \min_{X \text{ rank } k} \|A S_2^\mathsf{T} G_2 - A R_2^\mathsf{T} X U_2^\mathsf{T} S_2^\mathsf{T} G_2\|_{1,1}$$

*Proof.* The following fact is known:

**Fact 5.1** (Lemma F.1 from [1]). *Let $L$ be a $t$ dimensional subspace of $\mathbb{R}^s$. Let $C \in \mathbb{R}^{m \times s}$ be a matrix with $m = O\left(\frac{1}{\epsilon^2} t \log \frac{t}{\epsilon}\right)$ and i.i.d. standard Cauchy entries. With probability $99/100$, for all $x \in L$ we have*

$$(1 - \epsilon) \|x\|_1 \le \|Cx\|_{\mathrm{med}} \le (1 + \epsilon) \|x\|_1$$

The theorem statement is simply the lemma applied to $L = \operatorname{ColSpan}\left([A S_2^\mathsf{T} \mid A R_2^\mathsf{T}]\right)$. $\qquad \square$

## 5.2 Solving Small Instances

Given problems of the form $\hat{X} = \operatorname{argmin}_{X \text{ rank } k} \|Y - ZXW\|_{\mathrm{med},1}$, we leverage an algorithm for checking the feasibility of a system of polynomial inequalities as a black box.

**Lemma 5.1.** *[2] Given a set $K = \{\beta_1, \cdots, \beta_s\}$ of polynomials of degree $d$ in $k$ variables with coefficients in $\mathbb{R}$, the problem of deciding whether there exist $X_1, \cdots X_k \in \mathbb{R}$ for which $\beta_i(X_1, \cdots, X_k) \ge 0$ for all $i \in [s]$ can be solved deterministically with $(sd)^{O(k)}$ arithmetic operations over $\mathbb{R}$.*

**Theorem 5.2.** *Fix any $\epsilon \in (0, 1)$ and $k \in [0, \min(m_1, m_2)]$. Let $Y \in \mathbb{R}^{n \times m''}$, $Z \in \mathbb{R}^{n \times m_1}$, and $W \in \mathbb{R}^{m_2 \times m''}$ be any matrices. Let $C \in \mathbb{R}^{m' \times n}$ be a matrix of i.i.d. Cauchy random variables, and $G \in \mathbb{R}^{m'' \times m'' \operatorname{poly}(1/\epsilon)}$ be a matrix of scaled i.i.d. Gaussian random variables. Then conditioned on $C$ satisfying Fact 5.1 for the adjoined matrix $[Y, Z]$ and $G$ satisfying the condition of Fact 4.1, a rank-k projection matrix $X$ can be found that minimizes $\|C(Y - ZXW)G\|_{\mathrm{med},1}$ up to a $(1 + \epsilon)$-factor in time $\operatorname{poly}(m'm''/\epsilon)^{O(mk) + (m'' + m') \operatorname{poly}(1/\epsilon)}$, where $m = \max(m_1, m_2)$.*

We remark that if, as we do in our algorithm, we set the all the parameters $m$, $m'$ and $m''$ to be $\log \log d \sqrt{\log d} \cdot \operatorname{poly}(k/\epsilon)$, we can write the runtime of this step (Line 9 of Algorithm 2) as $(n + d) \operatorname{poly}(k/\epsilon) + \exp(\operatorname{poly}(k/\epsilon)))$. If $\operatorname{poly}(k/\epsilon) \le \sqrt{\log d}/(\log \log d)^2$, then this step is captured in the $(n + d) \operatorname{poly}(k/\epsilon)$ term. Otherwise this step is captured in the $\exp(\operatorname{poly}(k/\epsilon))$ term.

# 6 Experiments

In this section we empirically demonstrate the effectiveness of COARSEAPPROX compared to the truncated SVD. We experiment on synthetic and real world data sets. Since the algorithm is randomized, we run it 20 times and take the best performing run. For a fair comparison, we use an input sparsity time approximate SVD as in [4].

For the synthetic data, we use two example matrices all of dimension $1000 \times 100$. In Figure 1a we use a Rank-3 matrix with additional large outlier noise. First we sample $U$ random $100 \times 3$ matrix and $V$ random $3 \times 10$ matrix. Then we create a random sparse matrix $W$ with each entry nonzero with probability 0.9999 and then scaled by a uniform random variable between 0 and $10000 \cdot n$. We

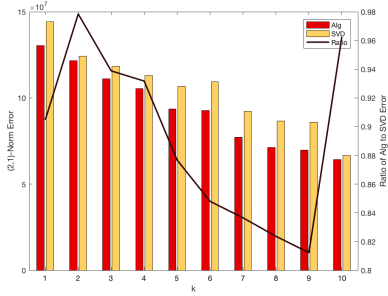

(a) Random Rank-3 Matrix Plus Large Outliers

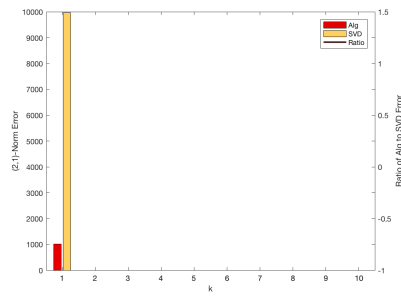

(b) Large Outlier Rank-2 Matrix

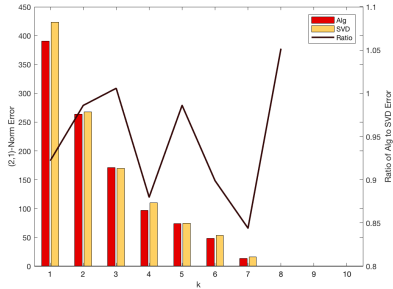

(c) Glass

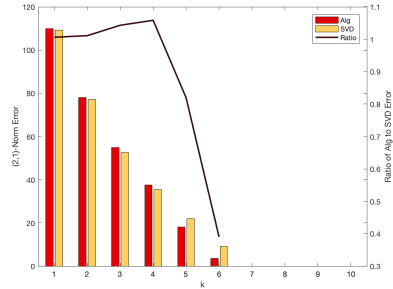

(d) E. Coli

Figure 1: Comparison of Algorithm 1 on synthetic and real world examples.

use $10 \cdot UV + W$. In Figure 1b we create a simple Rank-2 matrix with a large outlier. The first row is $n$ followed by all zeros. All subsequent rows are $0$ followed by all ones.

While the approximation guarantee of COARSEAPPROX is weak, we find that it performs well against the SVD baseline in practice on our examples, namely when the data has large outliers rows. The second example in particular serves as a good demonstration of the robustness of the (2,1)-norm to outliers in comparison to the Frobenius norm. When $k = 1$, the truncated SVD which is the Frobenius norm minimizer recovers the first row of large magnitude, whereas our algorithm recovers the subsequent rows. Note that both our algorithm and the SVD recover the matrix exactly when $k$ is greater than or equal to rank.

We have additionally compared our algorithm against the SVD on two real world datasets from the UCI Machine Learning Repository: Glass is a $214 \times 9$ matrix representing attributes of glass samples, and E.Coli is a $336 \times 7$ matrix representing attributes of various proteins. For this set of experiments, we use a heuristic extension of our algorithm that performs well in practice. After running COARSEAPPROX, we iterate solving $Y_t = \min_Y \|CAS^{\mathsf{T}}G - YZ_{t-1}\|_{1,1}$ and $Z_t = \min_Z \|CAS^{\mathsf{T}}G - Y_tZ\|_{1,1}$ (via Iteratively Reweighted Least Squares for speed). Finally we output the rank k Frobenius minimizer constrained to RowSpace($Y_tZ_t$). In Figure 1c we consistently outperform the SVD by between 5% and 15% for nearly all $k$, and nearly match the SVD otherwise. In Figure 1d we are worse than the SVD by no more than 5% for $k = 1$ to $4$, and beat the SVD by up to 50% for $k = 5$ and $6$. We have additionally implemented a greedy column selection algorithm which performs worse than the SVD on all of our datasets.

**Acknowledgements:** We would like to thank Ainesh Bakshi for many helpful discussions. D. Woodruff thanks partial support from the National Science Foundation under Grant No. CCF-1815840. Part of this work was also done while D. Woodruff was visiting the Simons Institute for the Theory of Computing.

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
