[Supplementary Material]

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

Applying this to all rows at once: $\|AX - A\|_{2,1} \in (1 \pm \epsilon) \|AXG - AG\|_{1,1}$.
We also use some existing machinery for input sparsity time $\ell_1$ subspace embeddings.

**Fact 4.1.1** (Theorem 4 from [19]). *For any given $D \in \mathbb{R}^{s \times t}$, let $\Pi \in \mathbb{R}^{r \times s}$ be a random Sparse Cauchy matrix with $r = O(t^5 \log^5 t)$ defined as follows: $\Pi = SC$ where $S \in \mathbb{R}^{r \times s}$ has each column uniformly and independently chosen from the $r$ standard basis vectors in $\mathbb{R}^r$, and where $C \in \mathbb{R}^{s \times s}$ is a diagonal matrix with diagonal entries chosen independently from the standard Cauchy distribution. Then with probability $99/100$ simultaneously for all $x \in \mathbb{R}^t$:*

$$\frac{1}{O(t^2 \log^2 t)} \cdot \|Dx\|_1 \leq \|\Pi Dx\|_1 \leq O(t \log t) \cdot \|Dx\|_1$$

**Fact 4.1.2** (Lemma D.25 from [23]). *If $\Pi \in \mathbb{R}^{r \times s}$ is a Sparse Cauchy matrix as defined above, and $B \in \mathbb{R}^{s \times t}$ is a fixed matrix, then with probability at least $99/100$:*

$$\|\Pi B\|_1 \leq O(\log(rt)) \|B\|_1$$

Finally, we also need a couple of structural lemmas which we state here without proof:

**Lemma 4.1.1** (Lemma 29 from [6]). *For a fixed $(B, D)$ pair such that $B \in \mathbb{R}^{r \times s}, D \in \mathbb{R}^{r \times t}$, if $S \in \mathbb{R}^{s/\operatorname{poly}(\epsilon) \times r}$ is a CountSketch Matrix composed with a matrix of i.i.d. Gaussians (for background on such sketching matrices, we refer the reader to the monograph [25]), then with probability $99/100$ both of the properties below hold:*

1. $\|S(BX - D)\|_{1,2} \geq (1 - \epsilon)\|BX - D\|_{1,2}$ *for any* $X$.
2. *If* $X^* = \mathrm{argmin}_{X \ rank \ k} \|BX - D\|_{1,2}$, *then* $\|S(BX^* - D)\|_{1,2} \leq (1 + \epsilon)\|BX^* - D\|_{1,2}$.

Clarkson and Woodruff [6] call such an $S$ a lopsided embedding for $(B, D)$ with respect to the $(1, 2)$-norm.

**Lemma 4.1.2** (Lemma 31 from [6]). *If* $R$ *is a lopsided embedding for* $(A_k^\mathsf{T}, A^\mathsf{T})$, *then:*

$$\min_{X \ rank \ k} \|AR^\mathsf{T}X - A\|_{2,1} \leq (1 + 3\epsilon)\Delta^*$$

Let $X' = \mathrm{argmin}_X \|TAR^\mathsf{T}X - TA\|_{2,1}$, $R^\mathsf{T} \in \mathbb{R}^{d \times \mathrm{poly}(k)}$ as in the lemma above and $\epsilon = O(1)$.

Define $E_1$ to be the event that the condition in Dvoretzky's theorem is satisfied, $E_2$ to be the event that Fact 4.1.1 holds for $D = AR^\mathsf{T}$, $E_3$ to be the event that Fact 4.1.2 holds for $B = AR^\mathsf{T}X^*G - AG$, and $E_4$ to be the event that $R$ satisfies Lemma 4.1.2.

$E_1$ holds w.h.p., $E_2, E_3, E_4$ each separately hold with probability $99/100$ (for a suitable choice of $K$). By a union bound, they all hold simultaneously with probability at least $24/25$. Conditioned on this happening:

$$\|AR^\mathsf{T}X' - A\|_{2,1} \leq \|AR^\mathsf{T}X^* - A\|_{2,1} + \|AR^\mathsf{T}(X^* - X')\|_{2,1} \tag{1}$$

$$\leq \|AR^\mathsf{T}X^* - A\|_{2,1} + \mathrm{poly}(k) \cdot \|TAR^\mathsf{T}(X^* - X')G\|_{1,1} \tag{2}$$

$$\leq \mathrm{poly}(k)\left(\|AR^\mathsf{T}X^* - A\|_{2,1} + \|T(AR^\mathsf{T}X^* - A)G\|_{1,1} + \|T(AR^\mathsf{T}X' - A)G\|_{1,1}\right) \tag{3}$$

$$\leq \mathrm{poly}(k)\left(\|AR^\mathsf{T}X^* - A\|_{2,1} + 2\|T(AR^\mathsf{T}X^* - A)G\|_{1,1}\right) \tag{4}$$

$$\leq \mathrm{poly}(k)\left(\|AR^\mathsf{T}X^* - A\|_{2,1} + O(\log d)\|(AR^\mathsf{T}X^* - A)G\|_{1,1}\right) \tag{5}$$

$$\leq \log d \cdot \mathrm{poly}(k)\|AR^\mathsf{T}X^* - A\|_{2,1} \tag{6}$$

(1) and (3) hold by the triangle inequality, (2) since $E_1$ and $E_2$ hold, (4) by $E_1$ again and since $X'$ is the minimizer of the expression $\|TAR^\mathsf{T}X - TA\|_{2,1}$, (5) since $E_3$ holds, and (6) by $E_1$ again.

$X'$ lies in the rowspace of $TA$, since otherwise there is a rank-$k$ projection $Z$ onto the rows of $TA$ with $\|TAX'Z - TAZ\|_{2,1} = \|TAX'Z - TA\|_{2,1}$ smaller than $\|TAX' - TA\|_{2,1}$. Since $E_4$ holds,

$$\|AR^\mathsf{T}X^* - A\|_{2,1} \leq O(1)\Delta^*$$

and thus the rowspace of $TA$ contains a $\log d \cdot \mathrm{poly}(k)$ approximation. $\square$

Thus the rowspace of $TA$ with $T$ as in Theorem 4.1 above is a $(\mathrm{poly}(k), \log d \cdot \mathrm{poly}(k))$-coreset for $A$.

## 4.2 Bootstrapping a Coreset

Given a poor coreset $Q$ for $A$, we now show how to leverage known results about residual sampling from [10] and [6] to obtain a better coreset of slightly larger dimension.

**Theorem 4.2.** *Given* $Q$, *an* $(\alpha, \beta)$-*coreset for* $A$, *with probability* $49/50$ BOOTSTRAPCORESET *returns an* $(\alpha + \beta\,\mathrm{poly}(k/\epsilon), (1 + \epsilon))$-*coreset for* $A$. *Furthermore* BOOTSTRAPCORESET *runs in space and time* $O(d\,\mathrm{poly}(\beta \log(nd)k/\epsilon))$, *with* $\mathrm{poly}(\beta \log(nd)k/\epsilon)$ *time per update in the streaming setting.*

**Algorithm 3** BOOTSTRAPCORESET

---

    **Input:** $A \in \mathbb{R}^{n \times d}$, $Q \in \mathbb{R}^{\alpha \times d}$ $(\alpha, \beta)$-coreset, $\epsilon \in (0, 1)$
    **Output:** $U \in \mathbb{R}^{(\alpha + \beta \operatorname{poly}(k/\epsilon)) \times d}$ $(\alpha + \beta \operatorname{poly}(k/\epsilon), (1 + \epsilon))$-coresets
  1: Compute $HA$ online // as in Lem. 4.2.2
  2: $P \leftarrow \beta \operatorname{poly}(k/\epsilon)$ samples of rows of $A(\mathcal{I} - Q)$ according to $\mathcal{P}(HA(\mathcal{I} - Q))$ // as in Lem. 4.2.2
  3: $U^{\mathsf{T}} \leftarrow$ Orthonormal basis for RowSpan $\left( \begin{bmatrix} Q \\ P \end{bmatrix} \right)$
  4: **return** $U^{\mathsf{T}}$

---

*Proof.* Consider the following idealized noisy sampling process that samples rows of a matrix $B$. Sample a row $B_i$ of $B$ with probability $\frac{\|B_i\|_2}{\|B\|_{2,1}}$ and add an arbitrary noise vector $E_i$ such that $\|E_i\|_2 \le \nu \|B_i\|_2$, where we fix the parameter $\nu = \frac{\epsilon}{100k\beta}$. Supposing we had such a process $\mathcal{P}^*(B)$, we can prove the following lemma.

**Lemma 4.2.1.** *Suppose $Q$ is an $(\alpha, \beta)$-coreset for $A$, and $P$ is a noisy subset of rows of the residual $A(\mathcal{I} - Q)$ of size $\beta(\operatorname{poly} k/\epsilon)$ each sampled according to $\mathcal{P}^*(A(\mathcal{I} - Q))$. Then with probability $99/100$, $\operatorname{RowSpan}(Q) \cup \operatorname{RowSpan}(P)$ is an $(\alpha + \beta \operatorname{poly}(k/\epsilon))$ dimensional subspace containing a $k$-dimensional subspace with corresponding projection matrix $X'$ such that:*

$$\left\| A - AX' \right\|_{2,1} \le (1 + \epsilon)\Delta^*$$

*Proof.* Our theorem is identical to Theorem 45 from [6], which is in turn an adaptation of Theorem 9 from [10], except that our sampling procedure produces noisy samples instead of actual rows of $A(\mathcal{I} - Q)$. We highlight the difference between our proof and the originals, and refer the reader to the sources for a full description.

Let $H_\ell$ denote the span of the rows of $Q$ adjoined with $\ell$ samples from $\mathcal{P}^*(A(\mathcal{I}-Q))$. The analysis considers $k + 1$ phases during the construction of $H_\ell$, where phase $j$ is defined such that there exists a subspace $X_j$ with:

  (i)  the dimension of $\operatorname{RowSpan}(X_j) \cap H_\ell \ge j$.

  (ii)  and letting $\delta = \epsilon/2k$ we have: $\|A(\mathcal{I} - X_j)\|_{2,1} \le (1 + \delta)^j \min_{X \text{ rank } k} \|A - AX\|_{2,1}$

In other words, the cost of the solution $X_j$ slowly gets worse with $j$, but $H_\ell$ recovers more of it. Note that in phase $k$, $\|A(\mathcal{I} - X_k)\|_{2,1} \le (1 + \epsilon) \min_{X \text{ rank } k} \|A - AX\|_{2,1}$, and furthermore $X_k \subseteq H_\ell$.

Let $Y_\ell$ denote the rank-$k$ projection whose row space is that of $X_j$, but rotated about the intersection $\operatorname{RowSpan}(X_j) \cap H_\ell$ such that it also contains the vector in $H_\ell$ realizing the smallest nonzero principle angle with $X_j$. Note that $Y_\ell$ satisfies condition (i) for some $j' > j$, so it remains to show that with high probability, with a small number of new samples, condition (ii) is also satisfied. In particular, we show that if condition (ii) is violated, and thus if:

$$\|A(\mathcal{I} - Y_\ell)\|_{2,1} > (1 + \delta) \|A(\mathcal{I} - X_j)\|_{2,1} \tag{1}$$

then with probability greater than $\delta/5K$ we sample a witness noisy-row $\hat{A}_{\ell'*}$ with the property:

$$\left\| \hat{A}_{\ell'*}(\mathcal{I} - Y_\ell) \right\|_2 \ge (1 + \delta/2) \left\| \hat{A}_{\ell'*}(\mathcal{I} - X_j) \right\|_2 \tag{2}$$

By the Angle Drop Lemma (Lemma 13 of [10]), this witness implies that the smallest nonzero principle angle between $X_j$ and $H_\ell$ (and so $Y_\ell$) decreases. By the analysis of Theorem 9 of [10], once the angle is small enough, $Y_\ell$ will satisfy (ii). We now prove this fact.

By the assumption on $\mathcal{P}^*$, $E_{\ell'}$ satisfies $\|E_{\ell'}\|_2 \leq \nu \|A_{\ell'*}(\mathcal{I} - Q)\|_2$. Recall we set the noise parameter $\nu = \frac{\epsilon}{100k\beta} = \frac{\delta}{50\beta}$.

Let $W$ denote the set of indices of witness *noisy* rows, in other words the set of all $i$ such that $\hat{A}_i$ satisfies (2). It suffices to show that:

$$\sum_{i \in W} \|A_{i*}(\mathcal{I} - Q)\|_2 \geq \frac{\delta}{5\beta} \|A(\mathcal{I} - Q)\|_{2,1} \tag{3}$$

Suppose that (3) is false. Let $\tilde{X}_\ell$ be the matrix projecting onto $H_\ell$.

$$\left\|\hat{A}_{i*}(\mathcal{I} - Y_\ell)\right\|_2 \leq \left\|\hat{A}_{i*}(\mathcal{I} - \tilde{X}_\ell)\right\|_2 + \left\|\hat{A}_{i*}\tilde{X}_\ell(\mathcal{I} - Y_\ell)\right\|_2 \tag{4}$$

$$\leq \left\|\hat{A}_{i*}(\mathcal{I} - \tilde{X}_\ell)\right\|_2 + \left\|\hat{A}_{i*}\tilde{X}_\ell(\mathcal{I} - X_j)\right\|_2 \tag{5}$$

$$\leq 2\left\|\hat{A}_{i*}(\mathcal{I} - \tilde{X}_\ell)\right\|_2 + \left\|\hat{A}_{i*}(\mathcal{I} - X_j)\right\|_2 \tag{6}$$

$$\leq 2\left\|\hat{A}_{i*}(\mathcal{I} - Q)\right\|_2 + \left\|\hat{A}_{i*}(\mathcal{I} - X_j)\right\|_2 \tag{7}$$

(4) and (6) follow from the triangle inequality, (5) since the definitions of $X_j, Y_\ell$ and $H_\ell$ imply that all elements of $H_\ell$ are closer to $\mathrm{RowSpan}(Y_\ell)$ than to $\mathrm{RowSpan}(X_j)$, and (7) since $\mathrm{RowSpan}(Q) \subseteq H_\ell$.

For $i \notin W$, by definition $\left\|\hat{A}_{i*}(\mathcal{I} - Y_\ell)\right\|_2 < (1 + \delta/2)\left\|\hat{A}_{i*}(\mathcal{I} - X_j)\right\|_2$. Combining both the bounds we have that for all $i$;

$$\left\|\hat{A}_{i*}(\mathcal{I} - Y_\ell)\right\|_2 \leq (1 + \delta/2)\left\|\hat{A}_{i*}(\mathcal{I} - X_j)\right\|_2 + [\![i \in W]\!] \cdot 2 \cdot \left\|\hat{A}_{i*}(\mathcal{I} - Q)\right\|_2$$

Summing over all $i$,

$$\left\|\hat{A}(\mathcal{I} - Y_\ell)\right\|_{2,1} \leq (1 + \delta/2)\left\|\hat{A}(\mathcal{I} - X_j)\right\|_{2,1} + 2\left\|\hat{A}_{W*}(\mathcal{I} - Q)\right\|_2$$

By triangle inequality:

$$\|A(\mathcal{I} - Y_\ell)\|_{2,1} - \|E(\mathcal{I} - Y_\ell)\|_{2,1} \leq \left[ \begin{array}{c} (1 + \delta/2)\|A(\mathcal{I} - X_j)\|_{2,1} + 2\|A_{W*}(\mathcal{I} - Q)\|_{2,1} \\ +(1 + \delta/2)\|E(\mathcal{I} - X_j)\|_{2,1} + 2\|E(\mathcal{I} - Q)\|_{2,1} \end{array} \right]$$

Finally, rearranging:

$$\|A(\mathcal{I} - Y_\ell)\|_{2,1} \leq \left(1 + \frac{\delta}{2}\right)\|A(\mathcal{I} - X_j)\|_{2,1} + \frac{2\delta}{5\beta} \cdot \beta \|A(\mathcal{I} - X_j)\|_{2,1} + 5\|E\|_{2,1} \tag{8}$$

$$\leq \left(1 + \frac{9\delta}{10} + 5\nu\beta\right)\|A(\mathcal{I} - X_j)\|_{2,1} \tag{9}$$

$$\leq (1 + \delta)\|A(\mathcal{I} - X_j)\|_{2,1}$$

Which contradicts our assumption that (1) held. (8) follows from the assumption that (3) is false and the fact that $\|A(\mathcal{I} - Q)\|_{2,1} \leq \beta \|A(\mathcal{I} - X_j)\|_{2,1}$ and (9) since $\|E\|_{2,1} \leq \nu \|A(\mathcal{I} - Q)\|_{2,1}$.

Note that this proof goes through for any error matrix $E$ satisfying $\|E_i\| \leq \nu \|A_i\|$ for all $i$. Also, as written in [6], the proof guarantees success with constant probability. We can repeat the sampling a constant number of times, keep all samples, and guarantee success with probability $99/100$. □

It remains to show that we can sample from $\mathcal{P}^*$ in a stream.

**Lemma 4.2.2.** *Let $B \in \mathbb{R}^{n \times d}$ be a matrix, and let $\delta, \nu \in (0,1)$ be given. Also let $s$ be a given integer. Then there is an oblivious sketching matrix $H \in \mathbb{R}^{\text{poly}(s/(\delta\nu)) \times n}$ and a sampling process $\mathcal{P}$, such that $\mathcal{P}(HB)$ returns a collection of $s' = O(s)$ distinct row indices $i_1, \ldots, i_{s'} \in [n]$ and approximations $\tilde{B}_{i_j} = B_{i_j} + E_{i_j}$ with $\|E_{i_j}\|_2 \leq \nu \cdot \|B_{i_j}\|_2$ for $j = 1, \ldots, s$. With probability $1 - \delta$ over the choice of $H$, the probability an index $i$ appears in the sampled set $\{i_1, \ldots, i_{s'}\}$ is at least the probability that $i$ appears in a set of $s$ samples without replacement from the distribution $\left( \frac{\|B_{1,*}\|_2}{\|B\|_{2,1}}, \ldots \frac{\|B_{n,*}\|_2}{\|B\|_{2,1}} \right)$. Furthermore the multiplication $HB$ and sampling process $\mathcal{P}$ can be done in $\text{nnz}(B) + d \cdot \text{poly}(s/(\delta\nu))$ time, and can be implemented in the streaming model with $d \cdot \text{poly}(s/(\delta\nu))$ bits of space.*

The theorem builds on the work of [1], [22] and [24].

*Proof.* We will show that with probability $O(1) \cdot \delta'$, we produce a set of $1 - O(1) \cdot s$ samples such that the probability a noisy row $\tilde{B}_i = B_i + E_i$ with $\|E_i\|_2 \leq \nu \|B_i\|_2$ appears in this set is at least $\frac{\|B_i\|_2}{\|B\|_{2,1}}$. Fixing $\delta' = \delta/O(1)$ will give the claim.

---

**Algorithm 4** H-SKETCH

**Input:** $B \in \mathbb{R}^{n \times d}$

**Output:** $HB \in \mathbb{R}^{d \, \text{poly}\left( \frac{s \log(nd)}{\nu\delta'} \right) \times d}$

1: **for** level $j \in [\ell]$ **do**
2:     $H_{j,*} \leftarrow$ new hash table with $w = O\left( \left( \frac{s \log n}{\nu\delta'} \right)^{15} \right)$ buckets and independent hash function $h_j \in ([n] \to [w])$ (each bucket stores a $d$ dimensional vector).
3:     Sample a set $J_j \subset [n]$ where each $i \in [n]$ is included with probability $p_j = \frac{1}{2^j}$.
4:     **for** $v \in [w]$ **do**
5:         $H_{j,v} = \sum_{i \in J_j} [\![ h_j(i) = v ]\!] \cdot \varepsilon_j(i) \cdot B_{i*}$ where $\varepsilon_j(i)$ are 2-wise ind. uniform $\pm 1$ random variables.
6:     **end for**
7: **end for**
8: **return** $\begin{bmatrix} H^{(1)} \\ H^{(2)} \\ \vdots \\ H^{(\ell)} \end{bmatrix}$

---

Before describing the algorithm we define a number of parameters.

- $M$ is an estimate for $\|B\|_{2,1}$ such that $\|B\|_{2,1} \leq M \leq O(1) \|B\|_{2,1}$ (we show in Appendix A.2 that we can calculate such an $M$ with high probability).

- Setting $T_j = M/2^j$, define $S_j = \{i \in [n] \mid \|B_i\|_2 \in (T_j, 2T_j]\}$ to be the $j$th level set of $B$.

- Define $s_j = |S_j|$ to be the number of rows in level $j$.

- For convenience, we also use the notation $S_{\geq j} = \bigcup_{j' \geq j} S_{j'}$ and $S_{\leq j} = \bigcup_{j' \leq j} S_{j'}$.

- Let $\ell = 4 \log(n/\delta)$ be the set of levels we consider in our sketch.

- Define a level $j \in [\ell]$ to be important if $s_j \geq \frac{\delta' 2^j}{\ell s}$. Informally, $j$ is an important level if the set of rows in in level $j$ contribute a significant fraction of $\|B\|_{2,1}$.

- Let $\mathcal{J} \subset [\ell]$ denote the set of all important levels.

Observe that for any level $j$ we have $s_j \leq 2^j$. It will suffice to consider only levels $j \in [\ell]$, since $s \leq n$ implies that these rows necessarily capture a $(1 - \delta'/s)$ fraction of the mass of $B$. By definition, the idealized process sampling from the distribution $\left( \frac{\|B_{1,*}\|_2}{\|B\|_{2,1}}, \dots \frac{\|B_{n,*}\|_2}{\|B\|_{2,1}} \right)$ will sample a level $j \in [\ell]$ with probability $1 - \delta'/s$, and by a union bound all $s$ samples come from such a level with probability $1 - \delta'$. Similarly, the idealized process will take a single sample from an important level with probability $(1 - \sum_{j \in [\ell]} T_j s_j / \|B\|_{2,1}) \geq 1 - \delta'/s$, meaning it only every samples from important levels also with probability $1 - \delta'$.

The main idea of the sketch is the following. For every level $j$, we subsample every row of $B$ independently at random with probability proportional to $2^{-j}$, and then hash the subsampled rows independently at random into buckets (each bucket is a vector that is the sum of the vectors assigned to it). Doing so guarantees that with high enough probability, for every important level $j$, there is a nearby level $k$ such that with high probability at least one row from $j$ is sampled in level $k$. Furthermore, all sufficiently heavy rows in level $k$ hash to different buckets, and all light rows contribute at most $\nu T_j$ to any one bucket. In particular, this means that if any bucket in important level $k$ has norm in the range $((1 - \nu)T_j, (2 + \nu)T_j]$, that bucket is of the form $\tilde{B}_i = B_i + E_i$ where $B_i$ is a row of $B$ and $E_i$ has small norm. We defer formal descriptions of these guarantees to Appendix A.1.

Next we argue that we can use the sketch from Algorithm 4 to produce samples from the idealized process with high enough probability. The general idea of the sampling algorithm SAMPLER is the following. Partition the rows of $B$ by assigning each row to one of $t = 100s^3$ pieces uniformly at random: $B^{(1)}$, $B^{(2)}$, ..., $B^{(t)}$. We can bound the probability that any two out of $s$ samples from the idealized process come from the same piece by $\binom{s}{2} \cdot \frac{\delta'}{100s^3} \leq \delta'$ so we can condition on this being the case. Sketch each piece using H-SKETCH to obtain: $H^{(1)}, H^{(2)}, ..., H^{(t)}$. Let $\ell^{(p)}, s_j^{(p)}, M^{(p)}, T_j^{(p)}$ denote the quantities $\ell, s_j, M$ and $T_j$ respectively for piece $B^{(p)}$. Using Lemma A.4.1, with constant probability we can calculate simultaneously for all $p$ an $O(1)$ estimate $\tilde{b}_p$ for $\|B^{(p)}\|_{2,1}$. Using Lemma A.3, we can also calculate simultaneously for all $j \in [\ell^{(p)}]$ a $O(1)$ estimate $\tilde{s}_j^{(p)}$ for $s_j^{(p)}$. Now repeat the following until we have generated $s'$ samples.

Sample a piece with probability proportional to $\tilde{b}_p$, and within that piece sample a level $j$ with probability proportional to $\tilde{s}_j T_j$. Examine level $j$ of the output of Algorithm 4. If at least one bucket of this level has a norm that is in the target range $(T_j, 2T_j]$, then output a uniform random choice of such a bucket. We show that this process generates samples with probabilities sufficiently close to those of the idealized process.

Let $C_i$ be the event that noisy row $\tilde{B}_i = B_i + E_i$ is extracted on line 7. Let $G_p$ be the event that piece $p$ is sampled on line 9. Let $D_j$ be the event that level $j$ is sampled on line 10. Finally let $E$ denote the event that any noisy row of the form $B_i + E_i$ with $\|E_i\|_2 \leq \nu \|B_i\|_2$ is extracted at all in iteration $z$. We wish to understand the probability of $C_i$:

$$\mathbb{P}\left[C_i\right] = \mathbb{P}\left[C_i \mid E \wedge D_{j'} \wedge G_{p'}\right] \cdot \mathbb{P}\left[E \mid D_{j'} \wedge G_{p'}\right] \cdot \mathbb{P}\left[D_{j'} \mid G_{p'}\right] \cdot \mathbb{P}\left[G_{p'}\right]$$

**Algorithm 5** SAMPLER

**Input:** $HB \in \mathbb{R}^{d\operatorname{poly}\left(\frac{s\log(nd)}{\nu\delta'}\right)\times d}$

**Output:** $B_{i_1}, \ldots, B_{i_{s'}}$ samples

1: Partition rows of $B$ uniformly at random into $t = 100s^3/\delta'$ pieces: $B^{(1)}, B^{(2)}, ..., B^{(t)}$.
2: **for** $p \in [t]$ **do**
3: $\quad$ $H^{(p)} \leftarrow$ H-SKETCH$(B^{(p)})$, computed online.
4: $\quad$ Calculate estimates $\tilde{b}_p \in \left[\left\|B^{(p)}\right\|_{2,1}, O(1) \cdot \left\|B^{(p)}\right\|_{2,1}\right]$ // as in Lemma A.4.1.
5: $\quad$ For all $j \in [\ell^{(p)}]$, calculate estimates $\tilde{s}_j^{(p)} \in \left[s_j^{(p)}, O(1) \cdot s_j^{(p)}\right]$ // as in Lemma A.3.
6: $\quad$ Set $M^{(p)} = \tilde{b}_p$.
7: **end for**
8: $F \leftarrow \emptyset$
9: **while** $|F| < s'$ **do**
10: $\quad$ Sample a piece $p' \in [t]$ with probability $\frac{\tilde{b}_{p'}}{\sum_p \tilde{b}_p}$ (without replacement).
11: $\quad$ Sample a level $j' \in [\ell]$ in $B^{(p')}$ with probability $\frac{\tilde{s}_{j'}^{(p')}T_{j'}^{(p')}}{\sum_j \tilde{s}_j^{(p')}T_j^{(p')}}$.
12: $\quad$ Let $k = \max\left(0, j' - 2\log\left(\frac{s\log n}{\delta'\nu}\right)\right)$.
13: $\quad$ **if** at least one bucket $v$ of $H_k^{(p')}$ has $\left\|H_{k,v}^{(p')}\right\|_2 \in ((1-\nu)T_{j'}, (2+\nu)T_{j'})$ **then**
14: $\quad\quad$ $F \leftarrow F \cup \left\{\text{uniform random } H_{k,v'}^{(p')} \text{ such that } \left\|H_{k,v'}^{(p')}\right\|_2 \in ((1-\nu)T_{j'}, (2+\nu)T_{j'})\right\}$
15: $\quad$ **end if**
16: **end while**
17: **return** $F$

---

We have straightforward bounds on the last two probabilities:

$$\mathbb{P}\left[G_p\right] = \frac{\tilde{b}_{p'}}{\sum_p \tilde{b}_p} = \Theta(1)\frac{\left\|B^{(p)}\right\|_{2,1}}{\|B\|_{2,1}}$$

$$\mathbb{P}\left[D_{j'} \mid G_p\right] = \frac{\tilde{s}_{j'}^{(p)}T_{j'}^{(p)}}{\sum_j \tilde{s}_j^{(p)}T_j^{(p)}} = \Theta(1) \cdot \frac{s_{j'}^{(p)}\|B_i\|_2}{\left\|B^{(p)}\right\|_{2,1}}$$

Now we can also lower bound $\mathbb{P}\left[E \mid D_j \wedge G_p\right]$. $E$ will not hold if either:

(i) a noisy row $\tilde{B}$ is sampled but $\tilde{B}$ cannot be written $B_i + E_i$ with $\|E_i\|_2 \leq \nu$.

(ii) no row is sampled at all.

If Lemmas A.1 and A.2 hold (i) will not occur. If Lemmas A.1 and Corollary A.1 hold (ii) will not occur. All these hold individually with probability at least $1 - O(1)/\log n$, so $E$ holds with probability at least $1 - O(1)/\log n$. Finally, since conditioned on $E \wedge D_{j'} \wedge G_{p'}$ we pick any row in level $j'$ from piece $p'$ with the same probability i.e. $1/s_j^{(p)}$. Putting all of this together, we get that:

$$\mathbb{P}\left[C_i\right] = \Theta(1) \cdot \frac{1}{s_j^{(p)}} \cdot \frac{s_{j'}^{(p)}\|B_i\|_2}{\left\|B^{(p)}\right\|_{2,1}} \cdot \frac{\left\|B^{(p)}\right\|_{2,1}}{\|B\|_{2,1}} = \Theta(1)\frac{\|B_i\|_2}{\|B\|_{2,1}}$$

To conclude, the sampling procedure samples noisy rows $B_i$ such that $i$ is sampled with probability at most a multiplicative constant from its probability under the distribution $\left( \frac{\|B_1\|_2}{\|B\|_{2,1}}, \dots \frac{\|B_n\|_2}{\|B\|_{2,1}} \right)$. Sampling $O(s)$ times guarantees that each row appears in the sampled set with at least the probability it would appear in $s$ samples of the idealized process.

Finally note that H-SKETCH, and the $\|\cdot\|_{2,1}$-norm estimation procedure of Lemma A.4.1, can be implemented as oblivious linear sketches. Since no two distinct pieces share any rows in common, all matrix multiplications can be done in $\mathrm{nnz}(B) + d \cdot \mathrm{poly}(s/(\delta\nu))$ time. Furthermore they can be implemented in the streaming model with $d \cdot \mathrm{poly}(s/(\delta\nu))$ bits of space. $\qquad\square$

Setting $b = \log(nd)$, $\delta = 1/100$, $\nu = \frac{\epsilon}{100k\beta}$ and $s = \beta \, \mathrm{poly}(k/\epsilon)$, it follows that $P$ contains $\beta \, \mathrm{poly}(k/\epsilon)$ samples from $\mathcal{P}^*(A(\mathcal{I}-Q))$ with probability $99/100$. By Lemma 4.2.1 and a union bound, the projection matrix of $\mathrm{RowSpan}(Q) \cup \mathrm{RowSpan}(P)$ is an $(\alpha + \beta \, \mathrm{poly}(k/\epsilon), (1+\epsilon))$-coreset for $A$ with probability $49/50$. BOOTSTRAPCORESET takes total time $O(\mathrm{nnz}(A)) + O(d \, \mathrm{poly}(\beta \log(nd)k/\epsilon))$ and space $O(d \, \mathrm{poly}(\beta \log(nd)k/\epsilon))$. $\qquad\square$

Note that in our main algorithm we cannot compute the projection $A(\mathcal{I}-Q)$ until the after the stream is finished. Fortunately, since $H$ is oblivious, we can right multiply $HA$ by $(\mathcal{I}-Q)$ once $Q$ is available, and only then perform the sampling procedure $\mathcal{P}$.

## 4.3 Right Dimension Reduction

We show how to reduce the right dimension of our problem. This result is used in both Algorithm 1 and Algorithm 2.

**Theorem 4.3.** *If $U^\mathsf{T}$ is an $(\alpha, \beta)$-coreset, $S \in \mathbb{R}^{\alpha \cdot \mathrm{poly}(k/\epsilon) \times d}$ is a CountSketch matrix composed with a matrix of i.i.d. Gaussians, and $R \in \mathbb{R}^{d \times \mathrm{poly}(k/\epsilon)}$ is a CountSketch matrix, then with probability $49/50$, if $X' = \mathrm{argmin}_X \|AS^\mathsf{T} - AR^\mathsf{T}XU^\mathsf{T}S^\mathsf{T}\|_{2,1}$ then:*

$$\left\|A - AR^\mathsf{T}X'U^\mathsf{T}\right\|_{2,1} \le (1 + O(\epsilon)) \min_{X \ rank \ k} \|A - AXU^\mathsf{T}\|_{2,1}$$

*Proof.* Here we apply reasoning similar to that at the bottom of page 32 from [6]. We need a couple of lemmas from [6].

**Lemma 4.3.1** (Lemma 30 from [6]). *If $S$ is a lopsided embedding for $(B, D)$, then if $X''$ has the property that $\|SBX'' - SD\|_{1,2} \le \kappa \min_{X \in \mathcal{C}} \|SBX - SD\|_{1,2}$ for some $\kappa$, then:*

$$\left\|BX'' - D\right\|_{1,2} \le \kappa(1 + 3\epsilon) \min_{X \in \mathcal{C}} \|BX - D\|_{1,2}$$

**Lemma 4.3.2.** *If $U \in \mathbb{R}^{d \times \alpha}$ and $R \in \mathbb{R}^{\mathrm{poly}(k/\epsilon) \times d}$ is a CountSketch matrix, then with probability $99/100$:*

$$\min_{X \ rank \ k} \|A - AR^\mathsf{T}XU^\mathsf{T}\|_{2,1} \le (1 + 3\epsilon) \min_{X \ rank \ k} \|A - AXU^\mathsf{T}\|_{2,1}$$

*Proof.* Let $V^* = \mathrm{argmin}_{V \ rank \ k} \|UV - A^\mathsf{T}\|_{1,2}$ and let $V = V_1 V_2$ be its rank factorization. Applying Lemmas 4.1.1 and 4.3.1, $R$ is a lopsided embedding for $(UV_1, A^\mathsf{T})$ with probability $99/100$. If $Y = \mathrm{argmin}_{Y \ rank \ k} \|R(UV_1 Y - A^\mathsf{T})\|_{1,2}$ then:

$$\|UV_1 Y - A^\mathsf{T}\|_{2,1} \le (1 + 3\epsilon) \|UV^* - A^\mathsf{T}\|_{1,2} \le (1 + 3\epsilon) \min_{X \ rank \ k} \|A - AXU^\mathsf{T}\|_{2,1}$$

But $Y = (RUV_1)^- RA^\mathsf{T}$, and taking transposes this means that:

$$\min_{X \text{ rank } k} \left\|A - AR^\mathsf{T}XU^\mathsf{T}\right\|_{2,1} \leq \left\|A - AR^\mathsf{T}((RUV_1)^-)^\mathsf{T}V_1^\mathsf{T}U^\mathsf{T}\right\|_{2,1} \leq (1+3\epsilon)\min_{X \text{ rank } k}\left\|A - AXU^\mathsf{T}\right\|_{2,1}$$

$\square$

From the last lemma, a solution to $\min_{X \text{ rank } k}\left\|A - AR^\mathsf{T}XU^\mathsf{T}\right\|_{2,1}$ will yield a $(1+O(\epsilon))$-approximate solution to the problem $\min_{X \text{ rank } k}\left\|A - AXU^\mathsf{T}\right\|_{2,1}$. Lemma 4.3.2 holds with probability $99/100$. Applying Lemma 4.1.1, with probability $99/100$, $S \in \mathbb{R}^{d \times \alpha \operatorname{poly}(k/\epsilon)}$ CountSketch composed with a Gaussian is a lopsided embedding for $(U, A^\mathsf{T})$. Union bounding over these events, and applying Lemma 4.3.1 with $\mathcal{C}$ as the set of matrices in $\operatorname{RowSpan}(RA^\mathsf{T})$ proves the claim with probability $49/50$. $\square$

## 4.4 Left Dimension Reduction

We show how to reduce the left dimension of our problem. Together with results from Section 4.3, this preserves the solution to $X^*$ to within a coarse $\sqrt{\log d}\log\log d \cdot \operatorname{poly}(k/\epsilon)$ factor.

**Theorem 4.4.** *Suppose the matrices $S_1$, $R_1$ and $U_1$ are as in Algorithm 1. If $C_1 \in \mathbb{R}^{\operatorname{poly}(k/\epsilon) \times n}$ is a Sparse Cauchy matrix, and $G_1 \in \mathbb{R}^{\log d \operatorname{poly}(k/\epsilon) \times \log d \operatorname{poly}(k/\epsilon)}$ is a matrix of appropriately scaled i.i.d. Gaussians (as in Fact 4.1), and*

$$\hat{X} = \operatorname*{argmin}_{X \text{ rank } k} \left\|C_1 AS_1^\mathsf{T}G_1 - C_1 AR_1^\mathsf{T}XU_1^\mathsf{T}S_1^\mathsf{T}G_1\right\|_F$$

*then with probability $24/25$:* $\left\|AS_1^\mathsf{T} - AR_1^\mathsf{T}\hat{X}U_1^\mathsf{T}S_1^\mathsf{T}\right\|_{2,1} \leq \sqrt{\log d}\log\log d \cdot \operatorname{poly}(k/\epsilon) \cdot \Delta^*$

*Proof.* Define $E_1$ to be the event that the condition in Dvoretzky's theorem is satisfied, $E_2$ to be the event that Fact 4.1.1 holds for $D = AR_1^\mathsf{T}$, and $E_3$ to be the event that Fact 4.1.2 holds for $B = (AS_1^\mathsf{T} - AR_1^\mathsf{T}X^*U_1^\mathsf{T}S_1^\mathsf{T})G_1$. $E_1$ holds w.h.p., $E_2$, $E_3$ each separately hold with probability $99/100$ (for a suitable choice of $K$). By a union bound, they all hold simultaneously with probability at least $24/25$. Conditioned on this happening:

$$\left\|AS_1^\mathsf{T} - AR_1^\mathsf{T}\hat{X}U_1^\mathsf{T}S_1^\mathsf{T}\right\|_{2,1} \leq \left\|AS_1^\mathsf{T} - AR_1^\mathsf{T}X^*U_1^\mathsf{T}S_1^\mathsf{T}\right\|_{2,1} + \left\|AR(X^* - \hat{X})U_1^\mathsf{T}S_1^\mathsf{T}\right\|_{2,1} \quad (1)$$

$$\leq \left\|AS_1^\mathsf{T} - AR_1^\mathsf{T}X^*U_1^\mathsf{T}S_1^\mathsf{T}\right\|_{2,1} + \operatorname{poly}(k/\epsilon)\left\|CAR(X^* - \hat{X})U_1^\mathsf{T}S_1^\mathsf{T}G_1\right\|_{1,1} \quad (2)$$

$$\leq \operatorname{poly}(k/\epsilon)\left[\begin{array}{l} \left\|AS_1^\mathsf{T} - AR_1^\mathsf{T}X^*U_1^\mathsf{T}S_1^\mathsf{T}\right\|_{2,1} + \left\|C(A - AR_1^\mathsf{T}X^*U_1^\mathsf{T})S_1^\mathsf{T}G_1\right\|_{1,1} \\ + \left\|C(A - AR_1^\mathsf{T}\hat{X}U_1^\mathsf{T})S_1^\mathsf{T}G_1\right\|_{1,1} \end{array}\right] \quad (3)$$

$$\leq \operatorname{poly}(k/\epsilon)\left[\begin{array}{l} \left\|AS_1^\mathsf{T} - AR_1^\mathsf{T}X^*U_1^\mathsf{T}S_1^\mathsf{T}\right\|_{2,1} + \left\|C(AS_1^\mathsf{T} - AR_1^\mathsf{T}X^*U_1^\mathsf{T}S_1^\mathsf{T})G_1\right\|_{1,1} \\ + \sqrt{\log d}\left\|C(A - AR_1^\mathsf{T}\hat{X}U_1^\mathsf{T})S_1^\mathsf{T}G_1\right\|_F \end{array}\right] \quad (4)$$

$$\leq \operatorname{poly}(k/\epsilon)\left[\left\|AS_1^\mathsf{T} - AR_1^\mathsf{T}X^*U_1^\mathsf{T}S_1^\mathsf{T}\right\|_{2,1} + \sqrt{\log d}\left\|C(AS_1^\mathsf{T} - AR_1^\mathsf{T}X^*U_1^\mathsf{T}S_1^\mathsf{T})G_1\right\|_{1,1}\right] \quad (5)$$

$$\leq \operatorname{poly}(k/\epsilon)\left[\begin{array}{l} \left\|AS_1^\mathsf{T} - AR_1^\mathsf{T}X^*U_1^\mathsf{T}S_1^\mathsf{T}\right\|_{2,1} \\ + \sqrt{\log d}\log\log d\left\|(AS_1^\mathsf{T} - AR_1^\mathsf{T}X^*U_1^\mathsf{T}S_1^\mathsf{T})G_1\right\|_{1,1} \end{array}\right] \quad (6)$$

$$\leq \sqrt{\log d}\log\log d\operatorname{poly}(k/\epsilon)\left\|AS_1^\mathsf{T} - AR_1^\mathsf{T}X^*U_1^\mathsf{T}S_1^\mathsf{T}\right\|_{2,1} \quad (7)$$

(1) and (3) hold by triangle inequality, (2) since $E_1$ and $E_2$ hold, (4) comes from the relationship between the 1-norm and 2-norm, (5) since $\hat{X}$ is the minimizer of the expression $\|C_1(A - C_1 AR_1^{\mathsf{T}} X U_1^{\mathsf{T}}) S_1^{\mathsf{T}} G_1\|_F$ and $p$-norms decrease with $p$, (6) since $E_3$ holds, and (7) by $E_1$ again. $\qquad\square$

The rank constrained Frobenius norm minimization problem above has a closed form solution.

**Fact 4.2.** *For a matrix $M$, let $U_M \Sigma_M V_M^{\mathsf{T}}$ be the SVD of $M$. Then:*

$$\operatorname*{argmin}_{X \; rank \; k} \|Y - ZXW\|_F = Z^- [U_Z U_Z^{\mathsf{T}} Y V_W V_W^{\mathsf{T}}]_k W^-$$

# 5 $(1 + \epsilon)$-Approximation

## 5.1 Left Dimension Reduction

The following median based embedding allows us to reduce the left dimension of our problem. Together with results from Section 4.3, this preserves the solution to $X^*$ to within a $(1 + O(\epsilon))$ factor.

**Theorem 5.1.** *Suppose $S_2$, $R_2$ and $U_2$ are as in Algorithm 2. If $C_2 \in \mathbb{R}^{\sqrt{\log d} \log\log d \, \mathrm{poly}(k/\epsilon) \times n}$ is a Cauchy matrix, and $G_2 \in \mathbb{R}^{\sqrt{\log d} \log\log d \, \mathrm{poly}(k/\epsilon) \times \sqrt{\log d} \log\log d \, \mathrm{poly}(k/\epsilon)}$ is a matrix of appropriately scaled i.i.d. Gaussians (as in Fact 4.1), and:*

$$\hat{X}' = \operatorname*{argmin}_{X \; rank \; k} \|C_2 A S_2^{\mathsf{T}} G_2 - C_2 A R_2^{\mathsf{T}} X U_2^{\mathsf{T}} S_2^{\mathsf{T}} G_2\|_{\mathrm{med},1}$$

*then with probability $99/100$:*

$$\left\| A S_2^{\mathsf{T}} G_2 - A R_2^{\mathsf{T}} \hat{X}' U_2^{\mathsf{T}} S_2^{\mathsf{T}} G_2 \right\|_{1,1} \le (1 + \epsilon) \min_{X \; rank \; k} \|A S_2^{\mathsf{T}} G_2 - A R_2^{\mathsf{T}} X U_2^{\mathsf{T}} S_2^{\mathsf{T}} G_2\|_{1,1}$$

*Proof.* The following fact is known:

**Fact 5.1** (Lemma F.1 from [2]). *Let $L$ be a $t$ dimensional subspace of $\mathbb{R}^s$. Let $C \in \mathbb{R}^{m \times s}$ be a matrix with $m = O\left(\frac{1}{\epsilon^2} t \log \frac{t}{\epsilon}\right)$ and i.i.d. standard Cauchy entries. With probability $99/100$, for all $x \in L$ we have*

$$(1 - \epsilon) \|x\|_1 \le \|Cx\|_{\mathrm{med}} \le (1 + \epsilon) \|x\|_1$$

The theorem statement is simply the lemma applied to $L = \mathrm{ColSpan}\left([A S_2^{\mathsf{T}} \mid A R_2^{\mathsf{T}}]\right)$. $\qquad\square$

## 5.2 Solving Small Instances

Given problems of the form $\hat{X} = \operatorname{argmin}_{X \; \mathrm{rank} \; k} \|Y - ZXW\|_{\mathrm{med},1}$, we leverage an algorithm for checking the feasibility of a system of polynomial inequalities as a black box.

**Lemma 5.1.** *[3] Given a set $K = \{\beta_1, \cdots, \beta_s\}$ of polynomials of degree $d$ in $k$ variables with coefficients in $\mathbb{R}$, the problem of deciding whether there exist $X_1, \cdots X_k \in \mathbb{R}$ for which $\beta_i(X_1, \cdots, X_k) \ge 0$ for all $i \in [s]$ can be solved deterministically with $(sd)^{O(k)}$ arithmetic operations over $\mathbb{R}$.*

**Theorem 5.2.** *Fix any $\epsilon \in (0, 1)$ and $k \in [0, \min(m_1, m_2)]$. Let $Y \in \mathbb{R}^{n \times m''}$, $Z \in \mathbb{R}^{n \times m_1}$, and $W \in \mathbb{R}^{m_2 \times m''}$ be any matrices. Let $C \in \mathbb{R}^{m' \times n}$ be a matrix of i.i.d. Cauchy random variables, and $G \in \mathbb{R}^{m'' \times m'' \, \mathrm{poly}(1/\epsilon)}$ be a matrix of scaled i.i.d. Gaussian random variables. Then conditioned on $C$ satisfying Fact 5.1 for the adjoined matrix $[Y, Z]$ and $G$ satisfying the condition of Fact 4.1, a rank-$k$ projection matrix $X$ can be found that minimizes $\|C(Y - ZXW)G\|_{\mathrm{med},1}$ up to a $(1 + \epsilon)$-factor in time $\mathrm{poly}(m'm''/\epsilon)^{O(mk) + (m'' + m') \, \mathrm{poly}(1/\epsilon)}$, where $m = \max(m_1, m_2)$.*

*Proof.* We write $X = PQ$, where $P$ is $m_1 \times k$ and $Q$ is $k \times m_2$, to ensure that $X$ is rank $\leq k$.

Guess a permutation $\pi_j$ for each column $j$ of $C(ZXW - Y)G$ and define constraints enforcing the permutation. Since the $(i, j)$-th entry of the matrix is $\sum_{k,\ell}(CZ)_{ik}X_{k\ell}(WG)_{\ell j} - (CYG)_{ij}$ these constraints are of the form $((C(ZXW - Y)G)_{\pi_j(i)j})^2 \leq ((C(ZXW - Y)G)_{\pi_j(i+1)j})^2$. Then define the median of the $j$-th column to be:

$$M_j = \left( |(C(ZXW - Y)G)_{\pi_j(\lfloor m''/2 \rfloor)j}| + |(C(ZXW - Y)G)_{\pi_j(\lceil m''/2 \rceil)j}| \right) / 2$$

which can be expressed via polynomial constraints. Thus we have $O(mk) + m'' \operatorname{poly}(1/\epsilon)$ variables in our polynomial inequality system, $O(mk)$ variables to describe $P$ and $Q$, and $m'' \operatorname{poly}(1/\epsilon)$ variables to describe the column medians $M_j$. We have $\operatorname{poly}(m'm''/\epsilon)$ constraints, each involving polynomials of $O(1)$ degree. By Lemma 5.1, checking the feasibility of this system takes time $\operatorname{poly}(m'm''/\epsilon)^{O(mk)+m'' \operatorname{poly}(1/\epsilon)}$. We can minimize the objective $\sum_j M_j$ using binary search. This requires a lower bound on the objective value, which we can get by noting from Fact 5.1 that:

$$\min_X \|CZXWG - CYG\|_{\mathrm{med},1} \geq (1 - \epsilon) \min_X \|ZXW - Y\|_{1,1} \geq (1 - \epsilon) \min_X \|ZXW - Y\|_{2,1}$$

As in the proof of Theorem 51 in [6], when the solution is constrained to be rank $k$, the right hand side is lower bounded by $\frac{1}{\operatorname{poly}(d)}(\sigma_{k+1}(Y))^{1/2}$ (where $\sigma_{k+1}(Y)$ is the $k+1$st singular value of $Y$), which itself is lower bounded by $\left( \frac{1}{\exp(\operatorname{poly}(m'm''))} \right)^k$. Thus we can do binary search in $\operatorname{poly}(m'm''/\epsilon)$ steps.

Finally, since there are $m'' \cdot m'!$ possible permutation guesses, the entire procedure takes time $\operatorname{poly}(m'm''/\epsilon)^{O(mk)+(m''+m')\operatorname{poly}(1/\epsilon)}$. $\qquad\square$

We remark that if, as we do in our algorithm, we set the all the parameters $m$, $m'$ and $m''$ to be $\log \log d \sqrt{\log d} \cdot \operatorname{poly}(k/\epsilon)$, we can write the runtime of this step (Line 9 of Algorithm 2) as $(n+d)\operatorname{poly}(k/\epsilon)+\exp(\operatorname{poly}(k/\epsilon)))$. If $\operatorname{poly}(k/\epsilon) \leq \sqrt{\log d}/(\log \log d)^2$, then this step is captured in the $(n+d)\operatorname{poly}(k/\epsilon)$ term. Otherwise this step is captured in the $\exp(\operatorname{poly}(k/\epsilon))$ term.

# 6 Experiments

In this section we empirically demonstrate the effectiveness of COARSEAPPROX compared to the truncated SVD. We experiment on synthetic and real world data sets. Since the algorithm is randomized, we run it 20 times and take the best performing run. For a fair comparison, we use an input sparsity time approximate SVD as in [5].

For the synthetic data, we use two example matrices all of dimension $1000 \times 100$. In Figure 1a we use a Rank-3 matrix with additional large outlier noise. First we sample $U$ random $100 \times 3$ matrix and $V$ random $3 \times 10$ matrix. Then we create a random sparse matrix $W$ with each entry nonzero with probability 0.9999 and then scaled by a uniform random variable between 0 and $10000 \cdot n$. We use $10 \cdot UV + W$. In Figure 1b we create a simple Rank-2 matrix with a large outlier. The first row is $n$ followed by all zeros. All subsequent rows are 0 followed by all ones.

(a) Random Rank-3 Matrix Plus Large Outliers

(b) Large Outlier Rank-2 Matrix

(c) Glass

(d) E. Coli

Figure 1: Comparison of Algorithm 1 on synthetic and real world examples.

While the approximation guarantee of COARSEAPPROX is weak, we find that it performs well against the SVD baseline in practice on our examples, namely when the data has large outliers rows. The second example in particular serves as a good demonstration of the robustness of the (2,1)-norm to outliers in comparison to the Frobenius norm. When $k = 1$, the truncated SVD which is the Frobenius norm minimizer recovers the first row of large magnitude, whereas our algorithm recovers the subsequent rows. Note that both our algorithm and the SVD recover the matrix exactly when $k$ is greater than or equal to rank.

We have additionally compared our algorithm against the SVD on two real world datasets from the UCI Machine Learning Repository: Glass is a $214 \times 9$ matrix representing attributes of glass samples, and E.Coli is a $336 \times 7$ matrix representing attributes of various proteins. For this set of experiments, we use a heuristic extension of our algorithm that performs well in practice. After running COARSEAPPROX, we iterate solving $Y_t = \min_Y \|CAS^\intercal G - YZ_{t-1}\|_{1,1}$ and $Z_t = \min_Z \|CAS^\intercal G - Y_t Z\|_{1,1}$ (via Iteratively Reweighted Least Squares for speed). Finally we output the rank k Frobenius minimizer constrained to RowSpace($Y_t Z_t$). In Figure 1c we consistently outperform the SVD by between 5% and 15% for nearly all $k$, and nearly match the SVD otherwise. In Figure 1d we are worse than the SVD by no more than 5% for $k = 1$ to 4, and beat the SVD by up to 50% for $k = 5$ and 6. We have additionally implemented a greedy column selection algorithm which performs worse than the SVD on all of our datasets.

**Acknowledgements:**   We would like to thank Ainesh Bakshi for many helpful discussions. D. Woodruff thanks partial support from the National Science Foundation under Grant No. CCF-1815840. Part of this work was also done while D. Woodruff was visiting the Simons Institute for the Theory of Computing.

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

# A  Elided Proofs

## A.1  H-SKETCH Guarantees

Note that in the proofs that follow, we require that our sketching matrices have small representations. For each level $j$ of each H-SKETCH, we can store the matrix as three different $O(1)$-wise independent hashes:

1. $P_j : [n] \to \{0, 1\}$ determines whether the row is subsampled
2. $h_j : [n] \to [w]$ determines the bucket
3. $\varepsilon_j : [n] \to \pm 1$ determines the sign.

Since our sketching matrices are stored with small seeds and do not guarantee full independence of the entries of the matrix, we cannot easily use Chernoff-Hoeffding bounds.

For ease of notation below, we define a parameter $C = O(\frac{s \log n}{\delta' \nu})$ with a sufficiently large constant.

For any level $j$, we separate the rows of $B$ into "light" and "heavy" rows with respect to $j$:

$$S_L^j = S_{\geq j+\log(C^6)+1}$$
$$S_H^j = S_{\leq j+\log(C^6)}$$

We first analyze the contribution of light and heavy elements to level $j$ of the sketch.

**Lemma A.1** (Noise from light elements is small). *For any $j \in [\ell]$, with probability $1 - O(1)/\log n$:*

$$\max_v \sum_{i \in S_L^j} [\![i \in I_j]\!][\![h_j(i) = v]\!] \cdot \|B_i\|_2 \leq \frac{0.1 \cdot \nu \cdot T_j}{C^2}$$

*Proof.* Define the variables $Z_i^v = [\![i \in J_j]\!] \cdot [\![h_j(i) = v]\!] \cdot \varepsilon_j(i) \cdot \|B_i\|_2$, where $i \in S_L^j$. For a fixed $v$, we first study $\sigma^2$, the second moment of $\sum_{i \in S_L^j} Z_i^v$.

$$\sigma^2 = \mathbb{E}\left[\left(\sum_i [\![i \in J_j]\!] \cdot [\![h_j(i) = v]\!] \cdot \varepsilon_j(i) \cdot \|B_i\|_2\right)^2\right] \tag{1}$$

$$= \sum_{i,i' \in S_L^j} \mathbb{E}\left[[\![i \in J_j]\!][\![i' \in J_j]\!] \cdot [\![h_j(i) = v]\!][\![h_j(i') = v]\!]\right] \cdot \mathbb{E}\left[\varepsilon_j(i)\varepsilon_j(i')\right] \cdot \|B_i\|_2 \cdot \|B_{i'}\|_2 \tag{2}$$

$$= \sum_{i \in S_L^j} \frac{p_j}{w} \cdot \|B_i\|_2^2 \tag{3}$$

$$\leq \frac{p_j}{w} \max_i(\|B_i\|_2) \cdot \sum_i \|B_i\|_2 \tag{4}$$

$$\leq \frac{T_j^2}{wC^6} \tag{5}$$

In step (2) we used the fact that the two variables $[\![i \in J_j]\!][\![i' \in J_j]\!][\![h_j(i) = v]\!][\![h_j(i') = v]\!]$ and $\varepsilon_j(i)\varepsilon_j(i')$ are independent. In step (3), we used the 2-wise independence of $\varepsilon_j$ and the fact that $\varepsilon_j(i)\varepsilon_j(i') = 1$ if $i = i'$ and 0 otherwise. In step (5) we used the fact that $\frac{p_j}{w}\|B\|_{2,1} \leq \frac{M}{w2^j} = \frac{T_j}{w}$ and:

$$\|B_i\|_2 \leq T_{j+\log(C^6)} = T_j/2^{\log(C^6)} = \frac{T_j}{C^6}$$

By Chebyshev's inequality:

$$\mathbb{P}\left[\sum_{i \in S_L^j} Z_i \geq \frac{0.1 \cdot \nu \cdot T_j}{C^2}\right] \leq \frac{O(1)}{w \log n}$$

The desired bound follows by a union bound over the $w$ buckets. □

**Lemma A.2** (Heavy Elements do not collide). *For any level $j \in [\ell]$, with probability at least $1 - \frac{O(1)}{\log n}$ no two elements from $S_H^j$ hash to the same bucket.*

*Proof.* We can bound the expected number of samples:

$$\mathbb{E}\left[\sum_{i \in S_H^j} [\![i \in I_j]\!]\right] \leq \left(\sum_{j'=1}^{j+\log(C^6)} s_{j'}\right) \cdot p_j \leq 2^{j+\log(C^6)+1} \cdot 2^{-j} = 2C^6$$

Thus, by Markov's bound,

$$\mathbb{P}\left[\sum_{i \in S_H^j} [\![i \in J_j]\!] > C^7\right] \leq \frac{O(1)}{\log n}$$

Thus, no more than $C^7$ heavy elements are subsampled with high probability. Conditioned on this happening, we can bound the probability that any two of them hash into the same bucket by:

$$w^{-1} \cdot \binom{C^7}{2} \leq \frac{C^{14}}{w} \leq \frac{1}{\log n}$$

since we chose $w = O(C^{15})$. The claim follows by a union bound over the two $1/\log n$ probability events. $\qquad\square$

**Lemma A.3** (Level estimates). *For any important level $j \in \mathcal{J}$, let $\hat{s}_j$ be the number of buckets in $H^{(k)}$ with norm in the interval $[(1-\nu)T_j, (2+\nu)T_j]$, where $k = \max\left(0, j - \log C^2\right)$. Let $\tilde{s}_j = 2\hat{s}_j p_k^{-1}$. Then with probability $1 - O(1)/\log n$ it holds that $\tilde{s}_j \in [s_j, 4s_j]$.*

*Proof.* By Lemma A.2, with probability $1 - O(1)/\log n$ the rows in $S_H^k$ do not collide, and by Lemma A.1, with probability $1 - O(1)/\log n$ the contribution of elements in $S_L^k$ is less than $\frac{0.1\nu T_k}{C^2} = 0.1\nu T_j$ (due to the relationship between $j$ and $k$). Conditioned on this holding, $\hat{s}_j = J_k^j$, where $J_k^j$ is the number of elements of $S_j$ subsampled in level $k$.

If $k = 0$, all rows of $S_j$ are subsampled, $p_k = 1$ and the claim is proved. Otherwise, we use a second moment method. $\mathbb{E}\left[J_k^j\right] = s_j p_k$ and additionally:

$$
\begin{aligned}
\mathbb{E}\left[\left|J_k^j\right|^2\right] &= \mathbb{E}\left[\left(\sum_{i \in S_j} [\![i \in J_k^j]\!]\right)^2\right] \\
&= \sum_{i,i' \in S_j} \mathbb{E}\left[[\![i \in J_k^j]\!][\![i' \in J_k^j]\!]\right] \\
&= \sum_{i \in S_j} \mathbb{E}\left[[\![i \in J_k^j]\!]\right] + \sum_{i \neq i' \in S_j} \mathbb{E}\left[[\![i \in J_k^j]\!][\![i' \in J_k^j]\!]\right] \\
&\leq s_j p_k + s_j^2 p_k^2
\end{aligned}
$$

Note that we only use the 2-wise independence of the subsampling function $P_k$. Thus $\left|J_k^j\right|$ has variance $\sigma^2 \leq s_j p_k$. Using Chebyshev's inequality:

$$\mathbb{P}\left[\left|\left|J_k^j\right| - s_j p_k\right| \geq \frac{s_j p_k}{2}\right] \leq \frac{2}{s_j p_k}$$

Since $s_j p_k \geq \log n$, the claim follows from a union bounding guaranteeing that Lemmas A.1 and A.2 hold together with the last event that the number of subsampled elements from $S_j$ is within a factor of two of its expectation. $\qquad\square$

**Corollary A.1** (At least one element is sampled). *For any important level $j \in \mathcal{J}$, with probability at least $1 - O(1)/\log n$, at least one element of $S_j$ is subsampled in $J_k$ when $k = \max\left(0, j - \log C^2\right)$.*

## A.2 High Probability $\|\cdot\|_{2,1}$ Estimation

**Lemma A.4.** *Given a matrix $B \in \mathbb{R}^{n \times d}$, there is an algorithm that with probability $1 - 1/(50t)$ outputs an estimate $M$ such that $\|B\|_{2,1} \le M \le 20010 \|B\|_{2,1}$. Furthermore this algorithm runs in time $O(\mathrm{nnz}(B) + O(n + d)(\mathrm{poly}\log(ndt)))$ and can be implemented in the streaming model with $d \cdot \mathrm{poly}(\log(ndt))$ bits of space.*

*Proof.* First we prove an intermediate result:

**Lemma A.4.1.** *If $S$ is a Count Sketch matrix with $O(t^2)$ rows, then with probability $1 - \frac{1}{100t}$, it holds that $\|B\|_{2,1}/2 \le \|BS^{\mathsf{T}}\|_{2,1} \le 2\|B\|_{2,1}$.*

*Proof.* For any fixed row $i$, if $S$ has $O(1/\delta)$ rows, then with probability at least $(1 - \delta)$ it holds that $\|B_i S^{\mathsf{T}}\|_2 \in (1 \pm 0.5)\|B_i\|_2$. For a proof of this, see e.g. Theorem 2.6 of [25] which shows that with probability $(1 - \delta)$, $S$ is a $(1 \pm 0.5)$ subspace embedding for $B_i^T$.

Let $T$ be the set of rows $i$ for which $\|B_i S^{\mathsf{T}}\|_2 \notin (1 \pm 0.5)\|B_i\|_{2,1}$. Then:

$$\|BS^{\mathsf{T}}\|_{2,1} = \sum_{i \in T} \|B_i S^{\mathsf{T}}\|_2 + \sum_{i \notin T} \|B_i S^{\mathsf{T}}\|_2$$

$$\le \sum_{i \in T} \|B_i S^{\mathsf{T}}\|_2 + \sum_{i \notin T} \frac{3}{2}\|B_i\|_2$$

Since $\mathbb{E}\left[\sum_{i \in T}\|B_i S^{\mathsf{T}}\|_2\right] \le \delta\|B\|_{2,1}$, by a Markov Bound:

$$\mathbb{P}\left[\sum_{i \in T}\|BS^{\mathsf{T}}\|_{2,1} \ge (3/2 + \gamma\delta)\|B\|_{2,1}\right] \le \frac{1}{\gamma}$$

For a lower bound, let $y_i = \begin{cases} 0 & \text{if } i \in T \\ \|B_i\|_2/2 & \text{if } i \notin T \end{cases}$, and let $z_i = \|B_i\|_2/2 - y_i$. Then $\mathbb{E}\left[\sum_i z_i\right] \le \frac{\delta\|B\|_{2,1}}{2}$.

Note that $\|BS^{\mathsf{T}}\|_{2,1} \ge \sum_{i \notin T}\frac{1}{2}\|B_i\|_2 = \sum_i y_i$. Again by a Markov Bound:

$$\mathbb{P}\left[\|BS^{\mathsf{T}}\|_{2,1} \le (1 - \gamma\delta)\|B\|_{2,1}\right] \le \mathbb{P}\left[\sum_i y_i \le (1 - \gamma\delta)\|B\|_{2,1}\right]$$

$$= \mathbb{P}\left[\sum_i z_i \ge \gamma\delta\|B\|_{2,1}\right]$$

$$\le \frac{1}{\gamma}$$

Setting $\gamma = 100t$ and $\delta = 1/10000t^2$, the claim is proved. $\qquad\square$

Calculating $BS^{\mathsf{T}}$ in a stream normally requires $\Omega(n)$ bits of space which exceeds our desired space bound. However we can remedy this with some lemmas from [24].

By Sections 4.1 and 4.2 of [24], for any matrix $B'$ with probability at least 0.9 it holds that:

$$\frac{2\|B'\|_{2,1}}{5} \le \left\|\mathrm{H\text{-}SKETCH}(B')\right\|_{2,1} \le 2001\|B'\|_{2,1}$$

for sufficiently large $\|B'\|_{2,1}$. Consequently, if we repeat this $O(\log t)$ times and let $M_0$ be the median of these trials, then $M_0$ achieves the same guarantee but with probability $1 - 1/(100t)$.

Letting $B' = B$ would naively require time $O(\text{nnz}(B') \cdot \log t)$, which exceeds our desired time bounds. However, using Lemma A.4.1, if we set $B' = BS^\mathsf{T}$ where $S$ is a Count Sketch matrix with $O(t^2)$ rows, we get a similar constant factor guarantee. By a union bound over the event that $M_0$ is a good estimator for $\|B'\|_{2,1} = \|BS^\mathsf{T}\|_{2,1}$ and the event that $\|BS^\mathsf{T}\|_{2,1}$ is itself a good estimator for $\|B\|_{2,1}$, with probability $1 - 1/(50t)$:

$$\frac{\|B\|_{2,1}}{5} \leq M_0 \leq 4002 \, \|B\|_{2,1}$$

Note that we can afford to store $O(\log t)$ copies of H-SKETCH($BS^\mathsf{T}$) in the stream. Outputting $M = 5 \cdot M_0$ yields the claim. $\qquad\square$