[Reviews · NeurIPS 2018]

Reviewer 1



This paper provides the first sublinear time algorithm for the subspace approximation problem in the streaming and in the distributed settings of computation. Subspace approximation is an important machine learning problem. The paper is very well written and i found the results interesting and important. One weakness of the paper is that the experimental results are short and not very informative.

Reviewer 2



This paper discusses the subspace estimation problem which is defined as follows: Given a set of n points in R^d and an integer k find a subspace S of dimension k such that the (loss function of) distance of the points to the subspace is minimised. There are (1+\eps)-approximation algorithm known for this problem with running time that is polynomial in n and d but exponential in (k/\eps). Moreover, these algorithms inherently do not work in the streaming setting where only one (or few) passes over the data is allowed. The main result claimed in this papers is a streaming (1+\eps)-approximation algorithm matching the running rime time bound of previous non-streaming algorithms. Streaming algorithms capture most realistic settings where the data is available only as a stream and does not sit in the memory. Given this, I consider the results in the paper significant. The paper also gives some experimental results on synthetic dataset. The main innovation in the paper is in the theoretical analysis which is not easy to follow for a non-expert. Many of the proofs is in the Supplementary material. The authors try to give some high-level ideas in Section 3. Unfortunately, much of the analysis is dependent on previously known results and it is hard for a non-expert to verify all these results quickly.

Reviewer 3



This paper studies the use of oblivious dimensionality reduction for approximating a point set with a subspace. It gives a sketching and solve algorithm that reduces both the number of points, and the dimension that they are in, to numbers close to the dimensional of the goal subspace. It then empirically demonstrates that the proposed methods perform better than SVDs. While subspace approximation is important, I’m doubtful of the value of this work for several reasons. First, unlike previous results on oblivious subspace embeddings that introduced new tools, this paper appears to be almost entirely applying existing tools to a mathematically natural variant of the problem. It does not discuss the connections and applications related to this problem. The experiments are also restricted on small synthetic data, and does not compare against natural heuristics such as simulated annealing and local methods. The experiment section also does not discuss the parameters that the projections are invoked with, and the role of table (b) is not very well explained. Overall, while this paper contains good theoretical ideas, it can benefit from a more thorough set of experiments on practically motivated datasets (although good steps in this direction were taken in the Author Feedback), as well as a more thorough treatment of the importance of the problem studied. On the other hand, due to the fundamental importance of subspace approximation problems in numerical linear algebra, I would not object to this paper appearing.